# Lethality of SARS-CoV-2 infection in K18 human angiotensin-converting enzyme 2 transgenic mice

Fatai S. Oladunni [1,3], Jun-Gyu Park [1,3], Paula A. Pino[1,3], Olga Gonzalez[1,3], Anwari Akhter[1,3], Anna Allué-Guardia[1,3], Angélica Olmo-Fontánez[1,2], Shalini Gautam[1], Andreu Garcia-Vilanova [1], Chengjin Ye[1], Kevin Chiem[1,2], Colwyn Headley [1], Varun Dwivedi[1], Laura M. Parodi[1], Kendra J. Alfson[1], Hilary M. Staples[1], Alyssa Schami[1,2], Juan I. Garcia[1], Alison Whigham[1], Roy Neal Platt II[1], Michal Gazi[1], Jesse Martinez[1], Colin Chuba[1], Stephanie Earley[1], Oscar H. Rodriguez[1], Stephanie Davis Mdaki[1], Katrina N. Kavelish[1], Renee Escalona[1], Cory R. A. Hallam[1], Corbett Christie[1], Jean L. Patterson [1], Tim J. C. Anderson[1], Ricardo Carrion Jr [1], Edward J. Dick Jr [1], Shannan Hall-Ursone[1], Larry S. Schlesinger[1], Xavier Alvarez[1], Deepak Kaushal [1], Luis D. Giavedoni[1], Joanne Turner [1✉], Luis Martinez-Sobrido [1✉] & Jordi B. Torrelles [1✉]

Vaccine and antiviral development against SARS-CoV-2 infection or COVID-19 disease would benefit from validated small animal models. Here, we show that transgenic mice expressing human angiotensin-converting enzyme 2 (hACE2) by the human cytokeratin 18 promoter (K18 hACE2) represent a susceptible rodent model. K18 hACE2 transgenic mice succumbed to SARS-CoV-2 infection by day 6, with virus detected in lung airway epithelium and brain. K18 ACE2 transgenic mice produced a modest TH1/2/17 cytokine storm in the lung and spleen that peaked by day 2, and an extended chemokine storm that was detected in both lungs and brain. This chemokine storm was also detected in the brain at day 6. K18 hACE2 transgenic mice are, therefore, highly susceptible to SARS-CoV-2 infection and represent a suitable animal model for the study of viral pathogenesis, and for identification and characterization of vaccines (prophylactic) and antivirals (therapeutics) for SARS-CoV-2 infection and associated severe COVID-19 disease.

---

[1] Texas Biomedical Research Institute, San Antonio, TX 78227, USA. [2] Integrated Biomedical Sciences Program, University of Texas Health Science Center at San Antonio, San Antonio, TX 78229, USA. [3]These authors contributed equally: Fatai S. Oladunni, Jun-Gyu Park, Paula A. Pino, Olga Gonzalez, Anwari Akhter, Anna Allué-Guardia. ✉email: joanneturner@txbiomed.org; lmartinez@txbiomed.org; jtorrelles@txbiomed.org

Human angiotensin-converting enzyme 2 (hACE2) protein is the functional receptor used by severe acute respiratory syndrome coronavirus 1 (SARS-CoV-1) to gain entry to cells[1,2]. Recently, hACE2 has also been described as the receptor for acute respiratory syndrome coronavirus 2 (SARS-CoV-2)[3–5], the etiological agent responsible for coronavirus disease 2019 (COVID-19). SARS-CoV-2 emerged in the city of Wuhan, China, in December 2019, causing a pandemic that has dramatically impacted public health and socioeconomic activities across the world[6–10]. Importantly, hACE2 is widely expressed in the lung, central nervous system, cardiovascular system, kidneys, gut, and adipose tissues where it negatively regulates the renin-angiotensin system, and facilitates amino-acid transport[5].

K18 hACE2 transgenic mice [B6.Cg-Tg(K18-ACE2)2Prlmn/J] are susceptible to SARS-CoV-1 infection[11] and recent reports suggest that K18 hACE2 transgenic mice can also be infected with SARS-CoV-2[12,13]. hACE2 expression in K18 hACE2 transgenic mice is driven by the human cytokeratin 18 (K18) promoter[11]. Importantly, hACE2 expression in K18 hACE2 transgenic mice is observed in airway epithelial cells where SARS-CoV-1 and SARS-CoV-2 infections are typically initiated. Recent research indicates that hACE2-expressing mice are useful for studies related to SARS-CoV-2 pathogenesis and COVID-19[12–16]. A validated rodent model of SARS-CoV-2 infection could help to accelerate testing of vaccines (prophylactic) and antivirals (therapeutic) for the prevention and treatment, respectively, of SARS-CoV-2 infection and associated severe COVID-19 disease. Compared with large animals, a murine model would have desirable features of tractability, ease of use and availability, be cost efficient and permit mechanistic studies to identify attributes of severe COVID-19 outcomes in some but not all people who are infected.

Transgenic mice expressing hACE2 have been developed using various promoters that produce mild-to-moderate SARS-CoV-2 infection in a variety of organs, in addition to physiological (weight loss, interstitial pneumonia) or immunological (anti-spike IgG) changes[12–16]. No transgenic mice models to date have led to SARS-CoV-2 infection-induced mortality[15]. Adenovirus-based delivery of hACE2 (Ad4-hACE2) to wild-type (WT) C57BL/6 mice resulted in susceptibility to SARS-CoV-2 infection but not mortality[13,15]. K18 hACE2 transgenic mice have previously been shown to represent a good animal model for SARS-CoV-1 infection and associated disease[11]. However, SARS-CoV-2 lethality in K18 hACE2 transgenic mice has not yet been fully determined.

In this study, we infected K18 hACE2 transgenic mice with SARS-CoV-2 to assess the feasibility of its use as an animal model of SARS-CoV-2 infection and associated COVID-19 disease. Contrary to other constitutively or transiently expressing hACE2 mouse models[12–19], K18 hACE2 transgenic mice were highly susceptible to SARS-CoV-2 infection, with all mice rapidly losing weight and succumbing to viral infection by 5–6 days post infection (DPI). Importantly, morbidity and mortality correlated with SARS-CoV-2 replication in the nasal turbinates, lungs, and brains at 2, 4, and 6 DPI. Notably, susceptibility was highly associated with a local and systemic chemokine storm, especially in the brain, and high levels of IFN-λ in the lungs by 6 DPI, with mild to moderate tissue pathology that included vasculitis, and the presence of SARS-CoV-2 nucleocapsid protein (NP) antigen and hACE2 expression in the nasal turbinates, lung epithelium and brain. In contrast, WT C57BL/6 mice survived viral infection with no changes in body weight and undetectable viral replication, NP antigen, and hACE2 expression. Altogether, our data provide evidence that K18 hACE2 transgenic mice represent an excellent animal model of SARS-CoV-2 infection and associated severe COVID-19 disease, providing the research community with a much-needed small animal model to evaluate vaccines and/or antivirals for SARS-CoV-2 infection and associated severe COVID-19 disease in vivo.

## Results

### K18 hACE2 transgenic mice are susceptible to SARS-CoV-2 infection.
K18 hACE2 transgenic and WT C57BL/6 mice where i. n. mock-infected (PBS), or infected with $1 \times 10^5$ plaque forming units, (PFU) of SARS-CoV-2, USA-WA1/2020 strain, and followed for body weight loss and survival for 14 days. By 6 DPI, SARS-CoV-2-infected K18 hACE2 transgenic mice lost over 20% of their initial body weight, were lethargic with rough fur and hunched appearance, immobile, and did not eat or drink, and succumbed to the infection (Fig. 1a, b). All K18 hACE2 transgenic mice succumbed to infection by 6 DPI, with the exception of one that was humanely killed due to IACUC-defined endpoints. Mock-infected K18 hACE2 transgenic or WT C57BL/6 mice, or SARS-CoV-2-infected WT C57BL/6 mice appeared healthy, maintained weight, and all survived SARS-CoV-2 infection for the duration of this study (Fig. 1a, b). Differentiating by sex, we observed that SARS-CoV-2-infected male K18 hACE2 transgenic mice lost weight starting 1 DPI, at the rate of ~5% per day whereas SARS-CoV-2-infected female K18 hACE2 transgenic mice did not begin to lose weight until 3 DPI (Supplementary Fig 1). In both cases, after 3-DPI, morbidity indicators accelerated until all K18 hACE2 transgenic mice, independent of sex, succumbed to SARS-CoV-2 infection by 6 DPI (Supplementary Fig 1).

To correlate body weight loss (Fig. 1a) and survival (Fig. 1b) with viral replication, K18 hACE2 transgenic and WT C57BL/6 mice were similarly mock-infected or infected with $1 \times 10^5$ PFU of SARS-CoV-2 and killed at 2-, 4-, or 6-DPI, to determine viral titers in the nasal turbinate, trachea, lung, heart, kidney, liver, spleen, small intestine, large intestine, and brain. By 2-DPI, K18 hACE2 transgenic mice had $\sim 1 \times 10^3$ PFU in nasal turbinate and $\sim 1 \times 10^5$ PFU in lung, indicating the presence of SARS-CoV-2 in both the upper and lower respiratory tracts. Viral titers at the nasal turbinates were maintained, but decreased in the lung by 4 DPI (Fig. 1c, d). SARS-CoV-2 was also detected in the brain at 4 DPI ($\sim 5 \times 10^5$ PFU) and increased further by 6-DPI ($1 \times 10^7$ PFU) (Fig. 1e), but was absent in all other organs. In all organs where SARS-CoV-2 was detected (nasal turbinate, lung, and brain), we observed no sex differences in viral titers (Supplementary Fig 2). SARS-CoV-2-infected WT C57BL/6 mice had undetectable viral loads at both DPI in all organs studied, in accordance with previous studies, suggesting that WT C57BL/6 mice are resistant to SARS-CoV-2 infection (Fig. 1a–e, and Supplementary Figs 1 and 2)[15]. Taken together, these results demonstrate the susceptibility of K18 hACE2 transgenic mice, but not WT C57BL/6 mice, to SARS-CoV-2 infection at the studied MOI, with detectable virus in the upper and lower respiratory tract, and brain, at early time points post-infection.

### SARS-CoV-2 infection of K18 hACE2 transgenic mice drives a local and systemic chemokine storm.
An early (2 DPI) chemokine storm was observed in the lungs of K18 hACE2 transgenic mice infected with SARS-CoV-2 (Fig. 2a). MIP-2/CXCL2, MCP-1/CCL2, MIP-1α/CCL3, MIP-1β/CCL4, RANTES/CCL5, and IP-10/CXCL10 levels were significantly increased in the lungs relative to C57BL/6 WT-infected mice or mock-infected K18 hACE2 transgenic mice. At 4 DPI several chemokines had decreased in magnitude but were significantly elevated when compared with C57BL/6 WT or mock-infected K18 hACE2 transgenic mice. For some chemokines, such as RANTES/CCL5, the high levels detected in the lung at 2 DPI were sustained at 4 DPI (Fig. 2a). Chemokine levels in the lungs of C57BL/6 WT mice did not

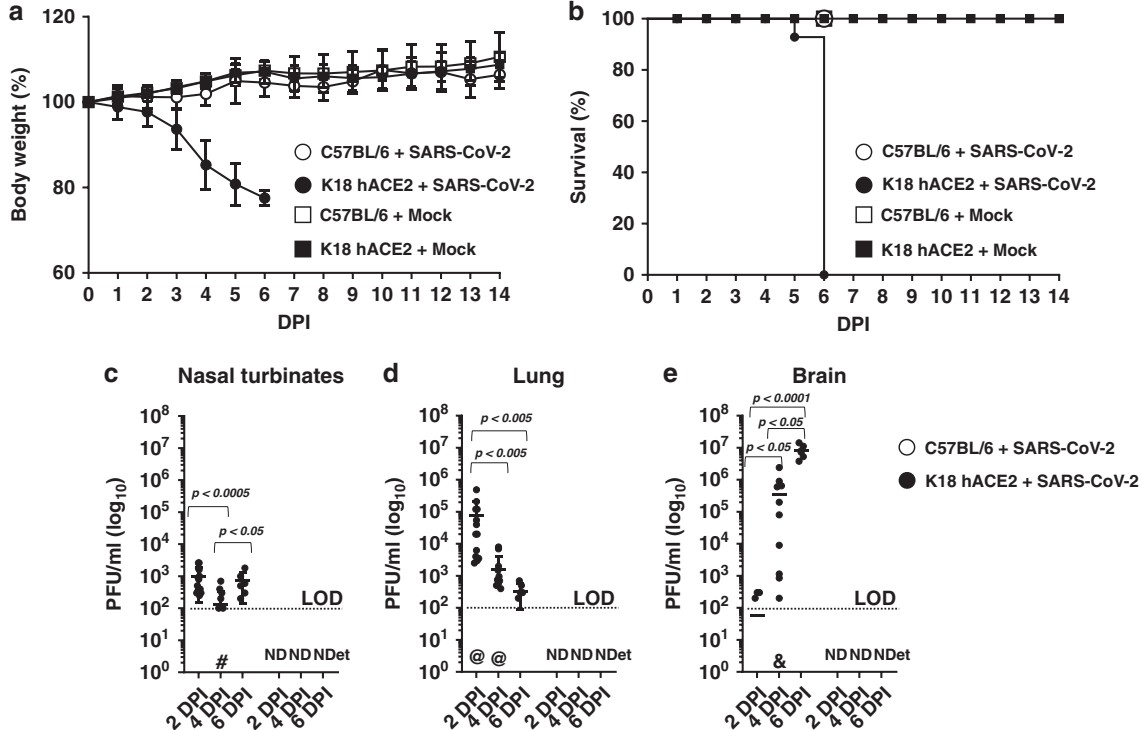

**Fig. 1 Infection of K18 hACE2 transgenic and WT C57BL/6 mice with SARS-CoV-2.** K18 hACE2 transgenic and WT C57BL/6 female and male mice were mock-infected or infected i.n. with $1 \times 10^5$ PFU of SARS-CoV-2. Body weight (**a**, $n = 14$ and $n = 6$ for K18 hACE2 transgenic mice infected and mock-infected respectively; $n = 8$ for WT infected and $n = 6$ for WT mock infected) and survival (**b**, $n$ values equal as in **a**) were evaluated at the indicated DPI. Mice that loss >25% of their initial body weight were humanely euthanized. Error bars represent standard deviations (SD) of the mean for each group of mice. **c–e** K18 transgenic hACE2 ($n = 34$ mice distributed as $n = 14$ for 2 DPI, $n = 14$ for 4 DPI and $n = 6$ for 6 DPI, 50:50 male:female) and WT C57BL/6 ($n = 16$ mice distributed as $n = 8$ for 2 DPI and $n = 8$ for 4 DPI) mice were similarly infected and killed at 2 and 4 DPI and viral titers in different organs (nasal turbinate, trachea, lung, brain, heart, kidney, liver, spleen, small intestine, and large intestine) were determined by plaque assay (PFU/ml). Data from virus-containing organs and/or tissue samples are shown: nasal turbinates **c**, lungs **d**, and brain **e**. Symbols represent data from individual mouse, and bars represent the geometric means of viral titers, $p < 0.05$; $p < 0.005$; $p < 0.0005$; $p < 0.0001$. @ virus not detected in one mouse, & virus not detected in four mice, # virus not detected in six mice, *ND* not detected, *NDet* non-determined. Dotted lines indicate the limit of detection, LOD ($10^2$ PFU/ml). *DPI* days post infection. Data are combined from two independent experiments. Source data are provided as a Source Data file.

increase in response to SARS-CoV-2 infection at either time point tested. A chemokine storm has been described in humans with COVID-19 disease, and it has been associated with development of acute respiratory distress syndrome (ARDS) in mice[20].

As a measure of systemic inflammation, we determined chemokine levels in the spleen, where MIP-2/CXCL2, MIP-1α/CCL3, MIP-1β/CCL4, and IP-10/CXCL10, were significantly increased at 2 DPI (Fig. 2b). Only MIP-1α/CCL3 and MIP-1β/CCL4 maintained a significant increase at 4 DPI when compared to both mock-infected K18 hACE2 transgenic mice and WT-infected C57BL/6 mice (Fig. 2b). Contrary to the lung, we did not observe an increase in RANTES/CCL5 relative to virus-infected C57BL/6 WT or mock-infected K18 hACE2 transgenic mice (Fig. 2b), identifying a potential sustained lung-specific chemokine response after SARS-CoV-2 infection. Our results indicate that SARS-CoV-2 infection of K18 hACE2 transgenic mice triggers both a local (lung) and systemic (spleen) chemokine storm. Chemokine analysis in nasal turbinate supported the early chemokine storm at 2 DPI in SARS-CoV-2-infected K18 hACE2 transgenic mice, with significant but transient high levels of MCP-1/CCL2, MIP-1α/CCL3, RANTES/CCL5, and IP-10/CXCL10 (Supplementary Fig 3a). No differences were observed in the trachea (Supplementary Fig 3b). Interestingly, the brain of SARS-CoV-2-infected K18 hACE2 transgenic mice also had significant levels of MIP-2/CXCL2, IP-10/CXCL10, and MIP-1α/CCL3 but in contrast to other organs, these were not detectable

until 4 DPI (Fig. 2c). We therefore determined chemokine levels in the brain at 6 DPI which were sustained at higher levels at 6 DPI (Supplementary Fig 5a), similar to the detection of virus in brain. There were no differences in chemokine levels in the lung between male and female K18 hACE2 transgenic-infected mice, with the exception of IP-10/CXCL10 at 2 DPI, which was significantly higher in female K18 hACE2 transgenic mice (Supplementary Fig 6a).

**SARS-CoV-2 infection of K18 hACE2 transgenic mice drives a local cytokine storm.** We next determined whether SARS-CoV-2 infection of K18 hACE2 transgenic mice resulted in a cytokine storm, as described during the development of severe COVID-19 disease in humans (reviewed in ref. [20–24]). Multiple tissues were analyzed to define local and systemic inflammatory, TH1, TH17, and TH2 responses. At 2 DPI the cytokine profile in the lungs of K18 hACE2 transgenic mice included a mixed inflammatory (TNF, IL-6, IFN-α, and IFN-λ), TH1 (IL-12, IFN-γ), TH17 (IL-17, IL-27) and TH2 (IL-4, IL-10) profile (Fig. 3a). Notably, IL-1β production was absent at 2- and 4 DPI. Reports in humans with COVID-19 showed that IL-1 precedes other cytokine production[24,25] Thus, studying earlier time points may have been informative in this mouse model with regard to IL-1β production.

Similar to that observed for chemokines, the cytokine storm resolved by 4 DPI with the exception of TNF and the type I and

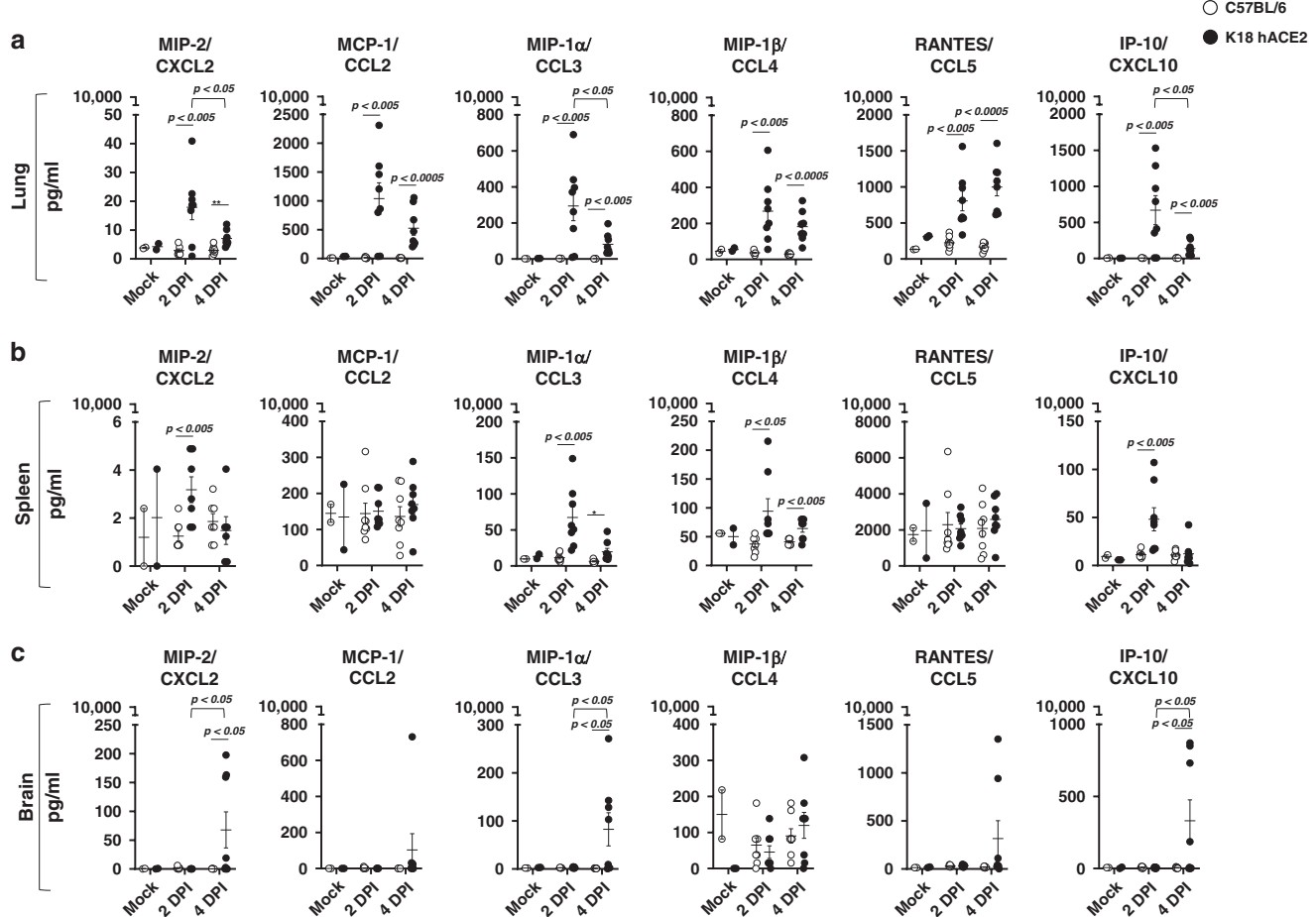

**Fig. 2 SARS-CoV-2-infected K18 hACE2 transgenic mice show a marked chemokine storm in selected tissues. a** Lung, **b** spleen, and **c** brain. Student's *t* test, two-tailed, C57BL/6 vs. K18 hACE2 $p < 0.05$; $p < 0.005$; $p < 0.0005$; two-WAY ANOVA C57BL/6 or K18 hACE2 transgenic mice over time, $p < 0.05$; $p < 0.005$; $p < 0.0005$, M ± SEM, $n = 8$ mice (50:50 male:female per time-point studied, except mock $n = 2$ mice). *DPI* days post infection. Data are representative over two independent experiments. Source data are provided as a Source Data file.

III IFNs, which remained significantly increased when compared with SARS-CoV-2-infected C57BL/6 WT or mock-infected K18 hACE2 transgenic mice (Fig. 3a). Reduction of chemokines and cytokines occurred despite detectable viral loads and extensive weight loss in virus-infected K18 hACE2 transgenic mice. No significant sex differences were found for cytokine levels in the lung.

With few exceptions, the cytokine storm was absent in the spleen (Fig. 3b), nasal turbinate (Supplementary Fig 4a), and trachea (Supplementary Fig 4b) of SARS-CoV-2-infected K18 hACE2 transgenic mice, indicating that the cytokine response was relatively localized to the lung. Exceptions included an increase in TNF at 2 DPI in the spleen (Fig. 3b), and a moderate but significant increase in IL-10 in the nasal turbinate at 2 DPI (Supplementary Fig 4a).

Interestingly, although we observed decreased cytokines in the lung by 4 DPI, some TH1 and TH17 cytokines (Fig. 3c) and chemokines (Fig. 2c) increased in the brain at this same time-point, suggesting a delayed viral spread to the brain. IL-1β, IL-27, IL-4, IL-13, and IL-10 were also elevated in the brain of C57BL/6 WT mice infected with SARS-CoV-2, even in the absence of detectable virus in any organs. IFN-α (type I) was only detected in the brain of SARS-CoV-2-infected K18 hACE2 transgenic by 6 DPI when compared with mock-infected controls (Supplementary Fig 5b). Both IFN-γ (type II) and IFN-λ (type III) were lower in K18 hACE2 transgenic mice relative to WT C57BL/6 mice

infected with SARS-CoV-2, and similar to mock-infected K18 hACE2 transgenic mice (Fig. 3c), except for 6 DPI where IFN-γ was significantly higher (Supplementary Fig 5b).

A clear sex difference was observed only in the brain. Males had significantly higher TH1 and TH17 cytokine responses at 4 DPI compared with SARS-CoV-2-infected female K18 hACE2 transgenic mice, and female mice had significantly higher TH2 responses when compared with male K18 hACE2 transgenic mice at the same time point (Supplementary Fig 6b). These sex differences are similar to studies of acute LPS-induced inflammation[26,27]. The TH1/TH2 sex difference was not observed in C57BL/6 WT mice infected with SARS-CoV-2 (Supplementary Fig 6b). Nonetheless, these differences between females and males normalized by 6-DPI, when mice succumbed to SARS-CoV-2 infection.

We further assessed correlations between viral titers, chemokines and cytokines observed in each tissue, and performed hierarchical clustering to identify the immune response characteristics in response to SARS-CoV-2 infection in K18 hACE2 transgenic mice (Fig. 4a–c and Supplementary Figs 7a, b). At 2 DPI in the lungs of SARS-CoV-2-infected K18 hACE2 transgenic mice we observed two distinct clusters (Fig. 4a): a cytokine cluster defined by the presence of IL-10 (Cluster-1, Fig. 4a), which correlated with the presence of IL-4 (0.94), IL-6 (0.91), IL-13 (0.91), and IL-12p70 (0.86); and a chemokine cluster (Cluster-2, Fig. 4a) defined by MCP-1/CCL2 and correlating with

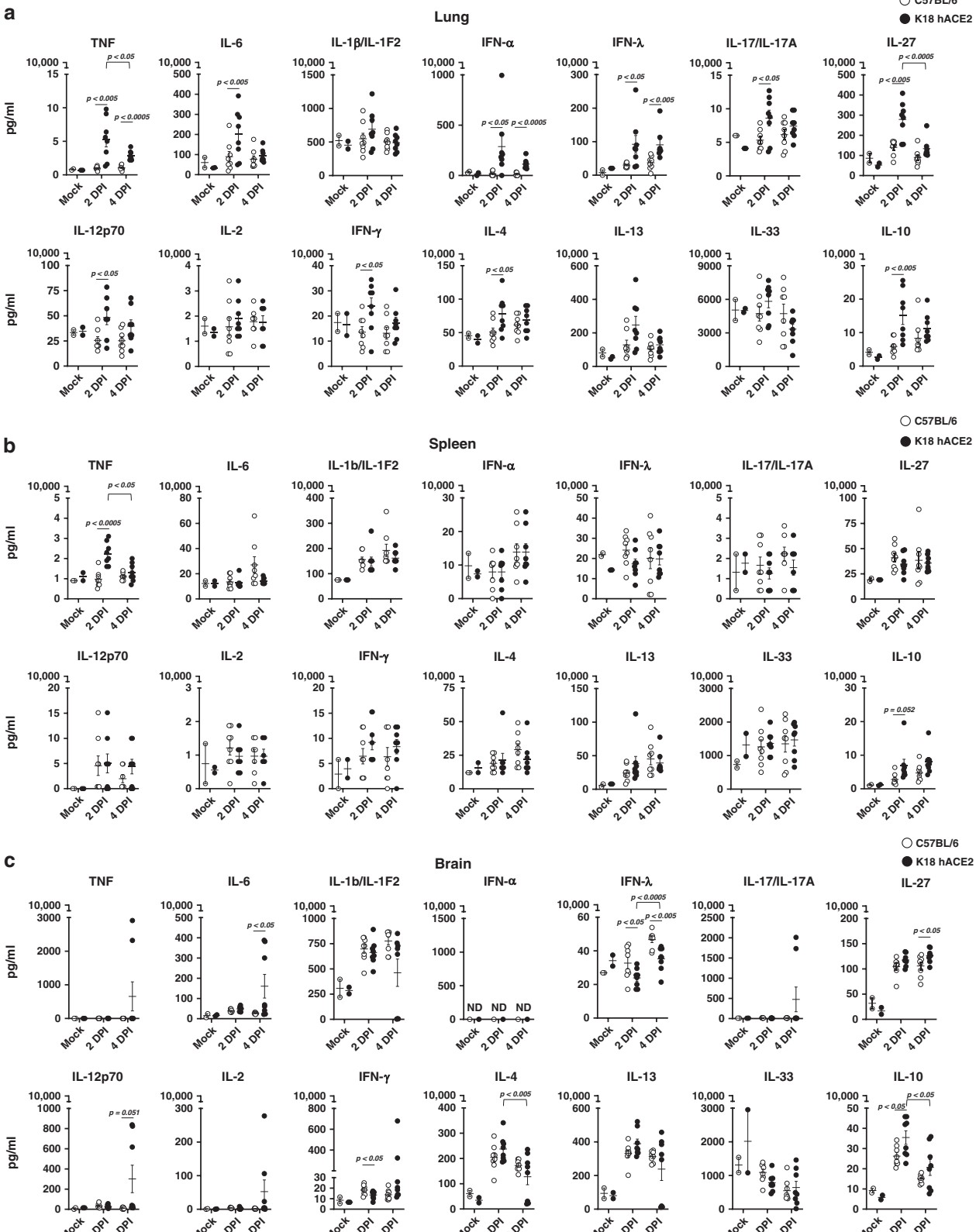

**Fig. 3 SARS-CoV-2-infected K18 hACE2 transgenic mice show a marked cytokine storm in selected tissues. a** Lungs, **b** spleen, and **c** brain. Student's *t* test, two-tailed, C57BL/6 vs. K18 hACE2 *p* < 0.05; *p* < 0.005; *p* < 0.0005; two-way ANOVA C57BL/6 or K18 hACE2 transgenic mice over time, *p* < 0.05; *p* < 0.005; *p* < 0.0005, M ± SEM, *n* = 8 mice (50:50 male:female per time-point studied, except mock *n* = 2 mice). *DPI* days post infection. Data are representative over two independent experiments. Source data are provided as a Source Data file.

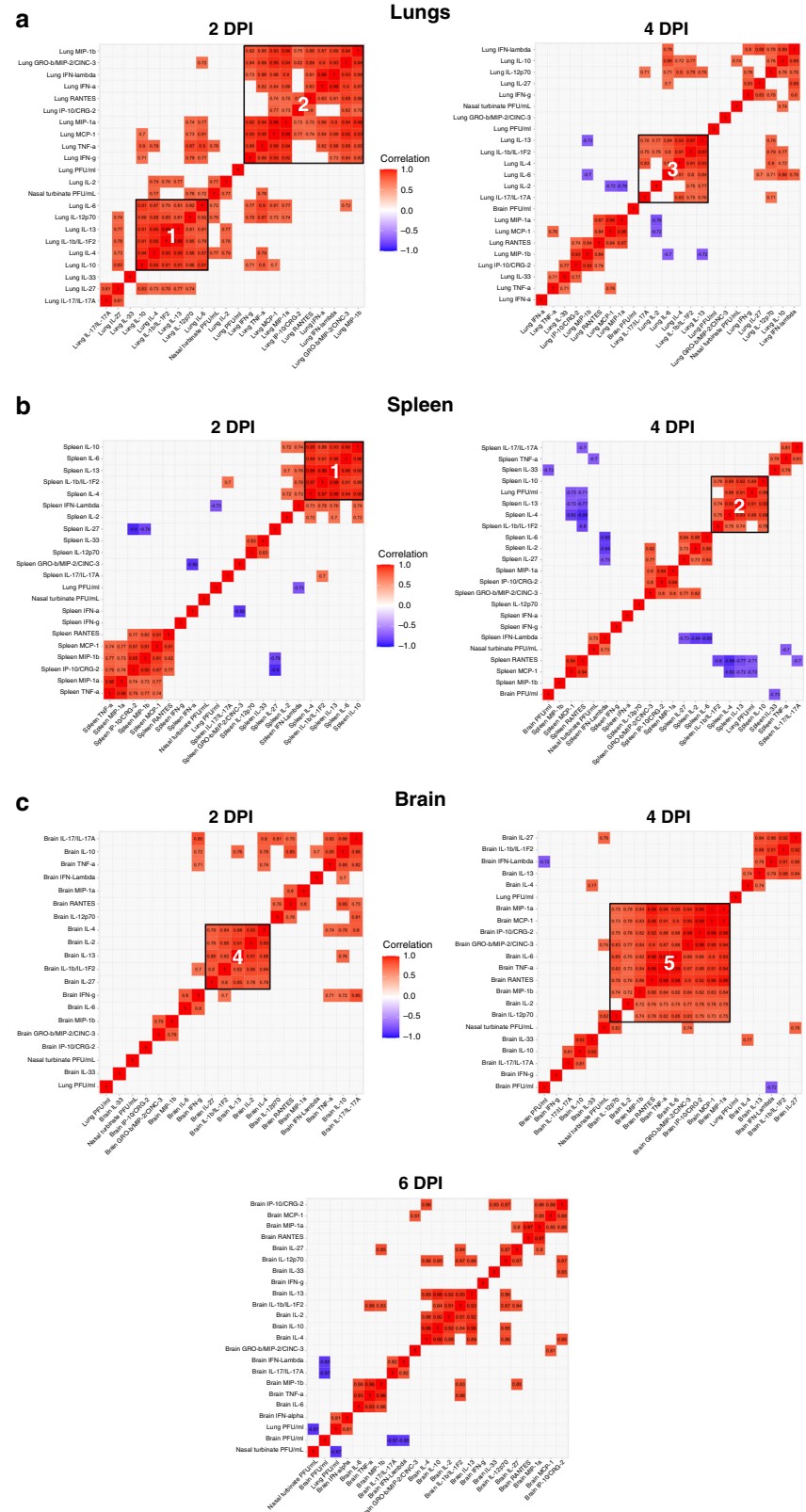

**Fig. 4 SARS-CoV-2-infected K18 hACE2 transgenic mice reveal differentiated clusters of chemokine and cytokine correlations with clinical symptoms progression.** Hierarchically clustered Pearson correlations of measurements in **a** lung, **b** spleen, and **c** brain of SARS-CoV-2-infected K18 hACE2 transgenic mice. Positive correlation (Red = 1) and negative correlations (Blue = −1), with clusters (Black outlined boxes with cluster number). Non-significant values ($p > 0.05$ measured by Pearson's correlation $t$ test) were left blank; $n = 8$ mice (50:50 male:female per time-point studied, except mock $n = 2$ mice). *DPI* days post infection. MIP-2/CXCL2; MCP-1/CCL2; MIP-1α/CCL3; MIP-1β/CCL4; RANTES/CCL5; IP-10/CXCL10. Data are representative over two independent experiments. Source data are provided as a Source Data file.

the presence of MIP-1β/CCL4 (0.99), MIP-2/CXCL2 (0.95) and MIP-1α/CCL3 (0.93), and cytokines TNF (0.95), and IFN-γ (0.93). Lung viral loads did not correlate with the presence of any specific cytokine or chemokine, possibly owing to the variability seen for viral loads and a quickly progressing disease. The 2 DPI correlative clusters in the lung were not maintained at 4 DPI but other smaller clusters were observed, the largest being led by IL-13 (Cluster-3, Fig. 4a) correlating with IL1-β (0.97) and IL-4 (0.93), suggestive of a rapidly developing immune response similar in nature, but not form, to the cytokine storm observed in humans[20–24].

The IL-10 cluster (Cluster-1) was also observed in the spleen at 2-DPI, correlating with IL-6 (0.96), IL-4 (0.95), IL-13 (0.93), and IL-1β (0.88); and at 4 DPI, correlating with IL-13 (0.92), IL-4 (0.89), and IL-1β (0.76)] (Fig. 4b), but not in the brain (Fig. 4c). Indeed, a correlation between IL-2, IL-4, and IL-13 was observed, independent of IL-10, in the brain at 2 DPI (Cluster-4, Fig. 4c), which was maintained band became IL-10 dependent by 6 DPI (Cluster-1, Fig. 4c). In the brain, a larger cluster (Cluster 5) was observed at 4 DPI, driven by RANTES/CCL5 and correlating with MCP-1/CCL2, MIP-1α/CCL3, TNF (0.99), and IL-6 (0.98) (Fig. 4c). Of these correlations only MCP-1/CCL2 and MIP-1α/ CCL3 were maintained by 6 DPI in the brain (Cluster 5, Fig. 4c). Nasal turbinates had a cytokine Cluster-6 that highly correlated with PFU in the lungs at both 2 DPI and 4 DPI (Supplementary Fig 7a). Other Clusters were identified in the trachea but these did not correlate with PFUs in any organ studied (Supplementary Fig 7b).

When analyzed by sex, only some differences were observed in the lung. Cluster-1 (Fig. 4a) had perfect correlations in males at 2 DPI but these correlations were absent in females with the exception of the IL-4 correlation with IL-10 (0.95) (Supplementary Fig 8a). Cluster-1 in males at 2 DPI did not extend to 4 DPI for either sex (Supplementary Fig 8b). Indeed, at 4 DPI in males, a new Cluster-9 with correlations between TNF, MCP-1/CCL2 and MIP-1α/CCL3 together with IL-12p70 and MIP-2/CXCL2 was observed (Supplementary Fig 8b).

In the brain, RANTES/CCL5, MIP-1α/CCL3, and IL-10 correlated with SARS-CoV-2 viral load in the nasal turbinate at 2 DPI only in males (Supplementary Fig 8c). This correlation disappeared at 4 DPI but a correlation between SARS-CoV-2 viral load and IL-10 in the brain was then observed (Supplementary Fig 8d). Conversely, SARS-CoV-2 viral load in the brain positively correlated with the presence of chemoattractants [MIP-2/CXCL2, IP-10/CXCL10, MCP-1/CCL2, and MIP-1α/ CCL3] in females (Supplementary Fig 8d). All correlated with the presence of RANTES/CCL5 and TNF in the brain, as well as with SARS-CoV-2 viral titers in nasal turbinate (Supplementary Fig 8d).

**SARS-CoV-2-infected K18 hACE2 transgenic mice develop rhinitis, pneumonia with associated pulmonary inflammation**. WT C57BL/6 mice developed minimal mononuclear and neutrophilic interstitial pneumonia (lung) (Fig. 5a, b, bracket) and rhinitis (nasal turbinate) that dissipated by 4 DPI (Fig. 5c, d), with very few consistent changes in other tissues with the exception of lymphocyte aggregates in the lamina propria of the small intestine (Fig. 5i, asterisk) and few small aggregates of mixed mononuclear inflammation with few neutrophils occasionally admixed with individual hepatocellular necrosis (Fig. 5j, arrowhead). Interestingly, these minor changes in gut and liver were independent of detectable virus, which was likely a consequence of the limit of detection in our assay, or to systemic inflammation. Minimal alveolar histiocytosis (Fig. 5d, asterisk), pneumocyte type II cells (Fig. 5d, arrowheads), perivascular mononuclear inflammation

(Fig. 5D, bracket) and rhinitis with low numbers of neutrophils (Fig. 5k, arrowhead) were variably observed in 4 DPI WT C57BL/ 6 mice (Fig. 5d). Brain tissue from SARS-CoV-2-infected C57BL/ 6 WT mice 4 DPI was normal (Fig. 5l). Thus, WT C57BL/6 mice had minimal to no pathologic findings consistent with their undetectable viral loads.

K18 hACE2 transgenic mice developed interstitial pneumonia (Fig. 5e, f) associated with alveolar histiocytosis admixed neutrophils and lymphocytes (Fig. 5f, asterisks), mild type II pneumocyte hyperplasia (Fig. 5f, arrowhead), bronchiolar syncytia (Fig. 5f, arrow), endothelial cell hyperplasia and vasculitis (Fig. 5f, bracket) by 2 DPI. Mice showed evidence of liver inflammation, although mixed inflammatory aggregates were minimal with variable amounts of individual hepatocellular necrosis (Fig. 5n, arrowhead), as well as gut-associated lymphoid tissue with prominent centers (Fig. 5m, asterisk). By 4 DPI, the majority of K18 hACE2 transgenic mice had 25% or greater lung involvement indicative of pneumonia (Fig. 5g), with affected areas presenting with inflammatory cellular accumulations and hemorrhage in alveolar spaces (Fig. 5h, asterisk) and interstitium (Fig. 5h, bracket), intra-alveolar fibrin admixed cellular debris (Fig. 5h, arrow), vasculitis (Fig. 5h, bracket), edema (Fig. 5h, arrowhead), and neutrophilic rhinitis (Fig. 5o, bracket). By 4 DPI, two K18 hACE2 transgenic mice had evidence of cerebral pathology; one animal had perivascular hemorrhage and another animal had mild meningoencephalitis with vasculitis (Fig. 5p, arrowhead). Summaries of histopathology for hACE2 transgenic mice infected with SARS-CoV-2 are described in detail in Tables 1 and 2.

We assessed the number of T (CD3[+]) and B (CD20[+]) cells present in the lung (Fig. 6) and brain (Fig. 7) of hACE2 transgenic mice (Table 3). Histiocytic cells and neutrophils were also quantified (Supplementary Fig 9). In the lungs, a modest migration of CD3[+] cells was observed at 2 DPI, relative to CD3[+] staining of lung tissue from non-infected mice, which was maintained at low levels until 6 DPI (when mice succumbed to SARS-CoV-2 infection). Histiocytes and neutrophils did not accumulate in the lung over time (Supplementary Fig 9). In contrast, we observed increasing neutrophils and CD3[+] cells in the brain as infection progressed (Supplementary Fig 9).

The overall severity score in the lungs of K18 hACE2 transgenic mice infected with SARS-CoV-2 (Supplementary Table 1) was minimal (≤10%) to mild (11–30%) at 2 DPI, which increased to moderate–mild (30–60%) by 4 DPI and to moderate–marked (60–80%) by 6 DPI. In the brain, K18 hACE2 transgenic mice maintained normality until 4 DPI, when a minimal score was observed in some mice; however, this score rapidly increased to mild-moderate by 6 DPI, before mice succumbed to SARS-CoV-2 infection.

**SARS-CoV-2 NP and hACE2 expression in tissues of SARS-CoV-2-infected K18 hACE2 transgenic mice**. Immunohistochemistry (IHC) labeling for SARS-CoV-2 NP antigen showed a heterogeneous distribution of viral NP in the lungs of K18 hACE2 transgenic mice; however, this distribution was more focalized in the nasal turbinate (Fig. 8a). Virus NP antigen was undetectable in all organs from C57BL/6 WT-infected mice (Fig. 8a). Distribution of the hACE2 receptor by IHC was obvious in the lungs and in the epithelium of the nasal turbinate of K18 hACE2 transgenic mice (Fig. 8a) and, as expected, absent from C57BL/6 WT mice (Fig. 8a). hACE2 was also highly expressed in the choroid plexus and some cells were also positive for SARS-CoV-2 NP (Fig. 8b). This result opens the possibility that the cerebral spinal fluid (CSF) produced by the choroid plexus, could

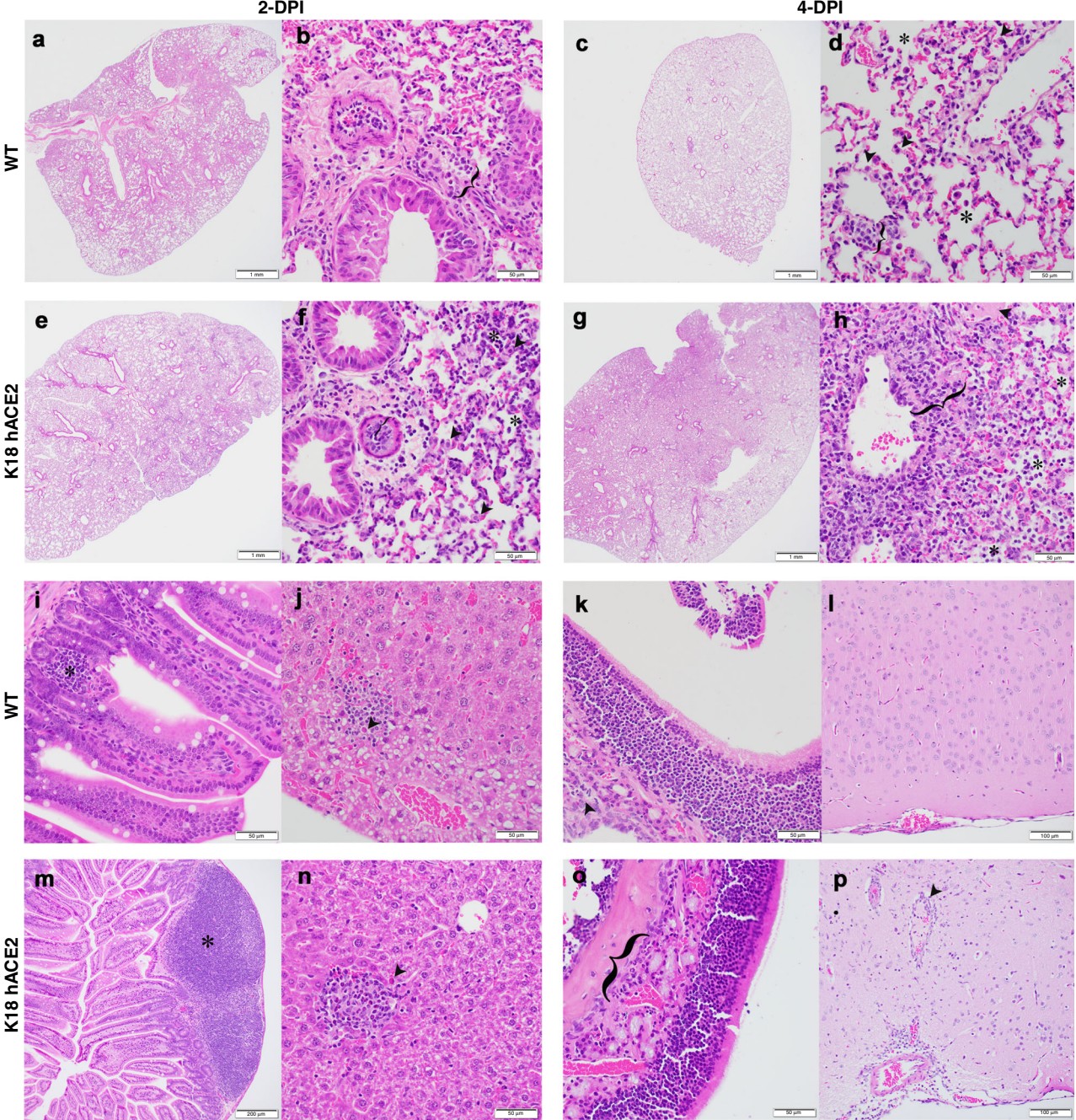

**Fig. 5 K18 hACE2 transgenic mice develop rhinitis, pneumonia with associated pulmonary inflammation after infection with SARS-CoV-2. a–d** and **i–l** WT C57BL/6 mice. Minimal mononuclear and neutrophilic interstitial pneumonia in WT lung at 2 DPI **a**, **b** bracket. By 4 DPI **c**, **d** minimal alveolar histiocytosis **d**, asterisk pneumocyte type II cells **d**, arrowheads, perivascular mononuclear inflammation **d**, bracket and rhinitis with low numbers of neutrophils **k**, arrowhead were variably observed. Lymphocyte aggregates in the lamina propria of the small intestine **i**, asterisk. Mixed mononuclear inflammation with individual hepatocellular necrosis **j**, arrowhead. Brain from WT 4 DPI was normal **l**. **e–h** and **m–p** K18 hACE2 transgenic mice. Interstitial pneumonia **e**, **f** associated with alveolar histiocytosis admixed neutrophils and lymphocytes **f**, asterisks, mild type II pneumocyte hyperplasia **f**, arrowhead, bronchiolar syncytia (**f**, arrow), endothelial cells hyperplasia and vasculitis (**f**, bracket) by 2 DPI. Gut-associated lymphoid tissue (GALT) with prominent germinal centers was observed (**m**, asterisk). Liver inflammation with variable amounts of individual hepatocellular necrosis (**n**, arrowhead). Greater lung involvement indicative of pneumonia (**g**), with inflammatory cellular accumulations and hemorrhage in alveolar spaces (**h**, asterisk) and interstitium (**h**, bracket), intra-alveolar fibrin admixed cellular debris (**h**, arrow), vasculitis (**h**, bracket), edema (**h**, arrowhead) by 4 DPI. Neutrophilic rhinitis observed at 4 DPI (**o**, bracket). Mild meningoencephalitis with vasculitis (**p**, arrowhead). Representative images of n = 8 (50:50 male:female per time-point and group) randomly chosen. Scale bars left images, 1 mm. Scale bars right images, 50 μm. DPI days post infection. Data are representative over two independent experiments.

**Table 1 Histopathology evaluation of SARS-CoV-2-infected WT and K18 hACE2 transgenic C57BL/6 mice.**

| Group | DAY 2 | | DAY 4 | |
|---|---|---|---|---|
| | Lung | Other | Lung | Other |
| C57BL/6 WT | ■ Minimal or mild multifocal random areas of interstitial thickening with mixed mononuclear cells and neutrophils and minimal intra-alveolar hemorrhage (4 F). ■ Rare vasculitis (1 F). ■ Multiple yet minimal perivascular mononuclear cells, low neutrophil numbers. ■ Rare or few bronchiolar syncytia (2 F/2 M). ■ Focal minimal aggregates of histiocytes. ■ Focal accumulation of amorphous eosinophilic material in alveolar space (fibrin) (1 M). | Nasal cavity: minimal to mild neutrophil infiltration in (1 F/1 M). Liver: minimal random multifocal aggregates of histiocytes, lymphocytes and few neutrophils. Rare necrotic hepatocytes within hepatic lobule (4 F/3 M). Small and large intestine: small focal aggregates of lymphocytes and rare scattered neutrophils in lamina propria (Gut-associated lymphoid tissue, GALT) (2 F). Kidney: focal small aggregate of neutrophils and histiocytes in interstitium of peri-renal adipose tissue, and medullary interstitium. Eosinophilic material in few glomeruli urinary spaces (1 M). Heart: small lymphoid aggregates within adipose tissue surrounding artery (1 F). | ■ Minimal perivascular and interstitial involvement. ■ Few neutrophils seen admixed with mononuclear inflammatory sites. ■ Occasional alveolar epithelial type II cells in areas of alveolar wall septal inflammation (4 F/2 M). ■ Rare vasculitis (1 F). ■ Few bronchiole syncytia (4 F/1 M). | Nasal cavity: in the majority, minimal or no changes. Large focal aggregate of lymphocytes and neutrophils in submucosa (1 F). Liver: minimal to mild random multifocal aggregates of histiocytes, lymphocytes and few neutrophils. Rare necrotic hepatocytes within hepatic lobule (3 F/3 M). Small and large intestine: multifocal minimal aggregates of lymphocytes within the lamina propria (GALT) (2 F/2 M). |
| K18 hACE2 | ■ Mild or moderate mononuclear and neutrophilic inflammation admixed with necrotic cell debris. ■ Predominant perivascular pattern expanding into alveolar septa. ■ Mild to moderate alveolar histiocytosis. ■ Mild alveolar epithelial cell type II hyperplasia. ■ Bronchiolar syncytia. ■ Endothelial syncytia and vasculitis. ■ Focal evidence of aspiration pneumonia. | Nasal cavity: few neutrophils within submucosa. Some lymphocyte, rare neutrophils (1 F/2 M). Liver: Minimal multifocal small aggregates of lymphocytes, histiocytes, neutrophils in hepatic cords (4 F/2 M). Small/Large intestine: Focal large/small lymphoid aggregates (GALT) (2 F/2 M). Few neutrophils in lamina propria (1 M). Kidney: Few focal areas of interstitial lymphocytes in deep cortex (1 F). Regional cystic dilation of renal pelvis (1 M). | ■ 25% or less (4 F/2 M) or >25% (2 M) of lung with dense mild to moderate mixed inflammatory infiltrate (macrophages, neutrophils) (3 F). ■ Regional consolidation mixed with necrotic debris. ■ Occasional intra-alveolar and perivascular hemorrhage. ■ Mild to moderate mixed mononuclear and neutrophilic inflammation. ■ Predominant interstitial, intra-alveolar and perivascular patterns of mixed inflammation. ■ Multifocal involvement of peribronchiolar, bronchiolar, and pleura in more severely affected areas. ■ Mild to moderate histiocytosis. ■ Mild pneumocyte type II hyperplasia. ■ Frequent vasculitis and endothelial syncytia. ■ Minimal intra-alveolar fibrin. ■ Bronchiolar syncytia present. ■ Regional perivascular edema admixed with mononuclear inflammation (1 M). | Nasal cavity: few foci of neutrophils, few lymphocytes (3 M). Low numbers of submucosal lymphocyte and plasma cell (1 M). Liver: Few focal aggregates of neutrophils, histiocytes, and lymphocytes (2 F/4 M). Small and large intestine: Focal small aggregate of lymphocytes or neutrophils in laminar propria (3 F/2 M). Brain: Multifocal perivascular hemorrhage (1 F). Multifocal perivascular lymphocytes, rare neutrophils, reactive microglial cells, neuronal necrosis and necrotic debris, vasculitis, inflammation extended to meninges (1 M). |

F female, M male. When not indicated, observed in the majority of females and males. Controls: WT 57BL/6 and K18 hACE2 transgenic mock-infected mice were assessed at the completion of the study, and showed no significant, or mild, microscopic lesions. Specifically, WT C57BL/6 mock-infected mice has no significant lesions, with the exception of prominent lymphoid follicles within the lamina propria (GALT) and splenic white pulp (1 M), or small lymphocyte aggregates admixed with few neutrophils in the hepatic sinusoids (1 F). K18 ACE2 transgenic mock-infected mice had minor findings; two small focal perivascular aggregates and focal osseous metaplasia within the pleura; few small aggregates of histiocytes and lymphocytes admixed with few neutrophils within hepatic sinusoids and occasionally in portal triads; prominent lymphoid follicles within gastrointestinal lamina propria (M); and macrophages within the splenic red pulp with minimal yellow intracytoplasmic pigment (1 F).

**Table 2 Histopathology evaluation of SARS-CoV-2-infected K18 hACE2 transgenic C57BL/6 mice in Lung and Brain at 6 DPI.**

| DAY 6 | Lung | Brain |
|---|---|---|
| K18 hACE2 | 5/6 showed moderate (20–60%) of the tissue had significant interstitial mixed neutrophilic and mononuclear inflammation expanding alveolar septal walls, prominently surrounding vascular structures and mildly filling alveolar spaces. Bronchioles were mostly spared of inflammation. 1/6 showed minimal interstitial mononuclear inflammation and occasionally rimming bronchiolar walls, airways were mostly empty and rare accumulation of pyknotic cellular debris noted in bronchiolar lumen and wall. Few alveolar spaces had minimal cellular debris admixed neutrophils and histiocytes. ~1–4% of the tissue involved. | 6/6 showed moderate (5–40%) mixed perivascular inflammation: mixed neutrophilic and mononuclear cell vasculitis in 2/6 or within the gray and white matter, within vascular walls and within meninges. (3/6). Areas of inflammation frequently had admixed necrotic cellular debris (5/6). Few neutrophils noted within the brain parenchyma interstitium (2/6). There were few individual glial cells and neuronal (hippocampus, or gray matter cortex) necrosis (3/6). Vasculitis was prominent (2/6). Some areas had swollen vacuolated neurons throughout gray matter (2/6). Few individual necrotic neurons within the hippocampus (1/6). |
| Diagnosis | Pneumonia (4/6), diffuse or multifocal, lymphoplasmacytic, histiocytic, interstitial, mild-moderate (5/6), or marked (1/6), subacute. | Meningoencephalitis (2/6) or encephalitis (4/6), neutrophilic, multifocal, mild, subacute, with necrosis, vasculitis (5/6). |

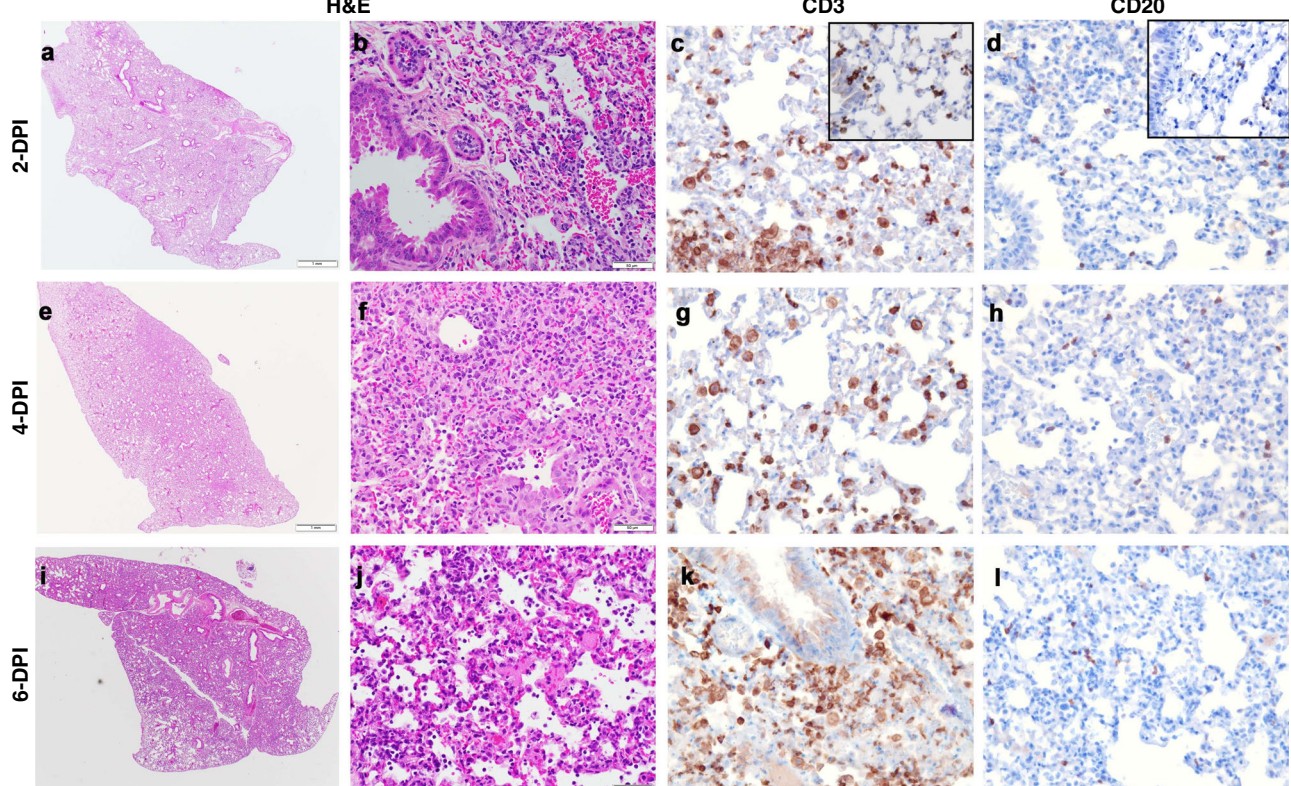

**Fig. 6 T- and B-cell accumulation in lung of SARS-CoV-2-infected K18 hACE2 transgenic mice.** Incremental inflammation was noted histologically at 2 (**a**, **b**), 4 (**e**, **f**), and 6 (**i**, **j**) DPI. There was significant CD3 cytoplasmic immunolabeling of interstitial round cells in the lung that were morphologically consistent with lymphocytes. Alveolar and interstitial macrophages in the lungs also had strong cytoplasmic immunolabeling with CD3. Evaluation of control tissues (inset) showed CD3 antibody clearly labeling other inflammatory cells that were morphologically not consistent with T cells **c**, **g**, **k**. Individual positive CD20 B cells were apparent in the interstitium of the lung alveolar septa. CD20 B cells were present in similar amounts in control (inset) SARS-CoV-2-infected tissues **d**, **h**, **l**. Representative images of n = 6 (50:50 male:female per time-point and group) randomly chosen. Scale bars left images, 1 mm. Scale bars right images, 50 μm. Data are representative over two independent experiments.

carry the virus into the central nervous system, disseminating SARS-CoV-2 through the body to the points of CSF reabsorption. Results were further confirmed by double staining (Fig. 8c–e) showing that co-localization of hACE2 with SARS-CoV-2 NP occurred primarily in the lungs, being more focalized in the nasal turbinates and brain. In the brain SARS-CoV-2 NP appeared in cell bodies of cortical neurons, a specific population of Pyramidal neurons in the cerebral cortex.

## Discussion

Herein we demonstrate that K18 hACE2 transgenic mice are highly susceptible to SARS-CoV-2 infection, quickly reaching study endpoints by 6-DPI following i.n. infection with $10^5$ PFU. SARS-CoV-2 was detected in the nasal turbinate and lung of K18 hACE2 transgenic mice on 2 and 4 DPI, and in the brain at 4 and 6 DPI. The presence of SARS-CoV-2 in the lung, nasal turbinate, and brain of infected hACE2 transgenic mice was verified by the

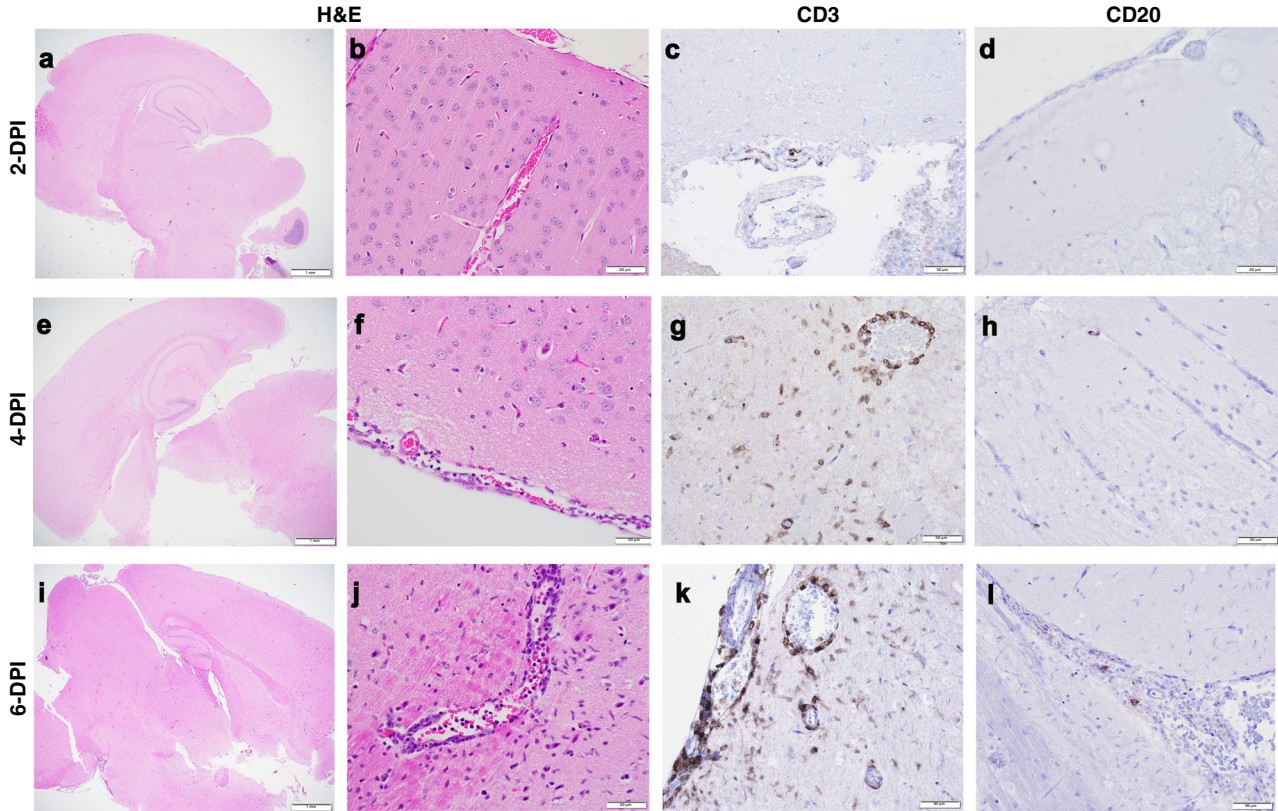

**Fig. 7 T- and B-cell accumulation in brain of SARS-CoV-2-infected K18 hACE2 transgenic mice.** Normal brain shown at 2 DPI (**a**, **b**). Incremental mononuclear infiltrates were noted histologically at 4 (**e**, **f**) and 6 (**i**, **j**) DPI. There was CD3 cytoplasmic immunolabeling of rare round cells morphologically consistent with lymphocytes in the brain 2 DPI (**c**). Incremental meningeal, intra-parenchymal, and perivascular CD3-positive cells were observed in the brain of 4 (**g**) and 6 (**k**) DPI. Brain of 2 (**d**) and 4 (**h**) DPI were mostly negative for CD20. CD20-positive cells were rare at 6 DPI (**l**). The outer margins of vascular structures and in some occasions the collagenous connective tissue of the meninges exhibited non-specific staining. The brain of 2 and 4 DPI mice had rare individual intra-parenchymal CD3 cells, some of which were not morphologically consistent with round cells. Tissue time points and staining as Fig. 6. Representative images of *n* = 6 (three males and three females per time-point and group) randomly chosen. Scale bars left images, 1 mm. Scale bars right images, 50 μm. Data are representative over two independent experiments.

---

**Table 3 T- and B-cell evaluation of SARS-CoV-2-infected K18 hACE2 transgenic C57BL/6 mice in lung and brain and severity scored.**

**T- and B-cell labeling in lung and brain**

Day 2 Lung: 6/6 had prominent CD3-positive round cells centered on and adjacent to bronchioles and surrounding vascular structures with lesser numbers extending into alveolar interstitium and alveolar spaces. 6/6 had rare intra-bronchiolar and intra-alveolar round cells with strong CD3 cytoplasmic staining. Semi-quantitative assessment of CD3-positive round cells correlates with the H&E inflammation scores; 5/6 had few/scant CD20-positive round cells with uniform distribution throughout alveolar septa and 1/6 had slightly increased CD20-positive round cells surrounding vascular structures.

Brain: 3/6 with rare intra-parenchymal and perivascular CD3 immunostaining; 2/6 with single/sparse CD3-positive cells located intra-parenchymal, ependymal, interstitium, and perivascular; CD20 negative. 5/6 CD20 negative.

Day 4 Lung: 4/6 had diffuse and 2/6 regional/multifocal CD3-positive round cells within alveolar septa, vascular structures, and bronchiolar walls. 6/6 had individual intra-bronchiolar and intra-alveolar CD3-positive round cells with strong cytoplasmic staining. Semi-quantitative assessment of CD3-positive round cells correlates with the H&E inflammation scores; 4/6 CD20-positive round cells had few/scant round cells within the interstitium with a uniform distribution and 2/6 had slightly increased CD20 round cells within areas of interstitial inflammation.

Brain: 5/6 Few or sparse CD3-positive cells located intra-parenchymal and perivascular or interstitium of ependyma and rimming vascular structures; 5/6 CD20 negative; 1/6 perivascular CD20-positive cells.

Day 6 Lung: 6/6 had prominent CD3-positive round cells within alveolar septa, intra-alveolar spaces and vascular walls. 4/6 had diffuse/uniform or 2/6 regional distribution of CD3-positive round cells throughout the interstitium with occasional small aggregates expanding the alveolar septa or vascular adventitia. 6/6 had rare intra-bronchiolar and intra-alveolar CD3-positive cells with strong cytoplasmic staining. Semi-quantitative assessment of CD3-positive round cells correlates with the H&E inflammation scores; 6/6 had few/scant CD20-positive round cells with uniform distribution throughout alveolar septa.

Brain: 5/6 significant/mild-marked intra-parenchymal and perivascular CD3-positive cells: 3/6 with moderate to marked meningeal CD3-positive cells. 5/6 CD20 negative or rare.

When not indicated, observed in the majority of females and males.

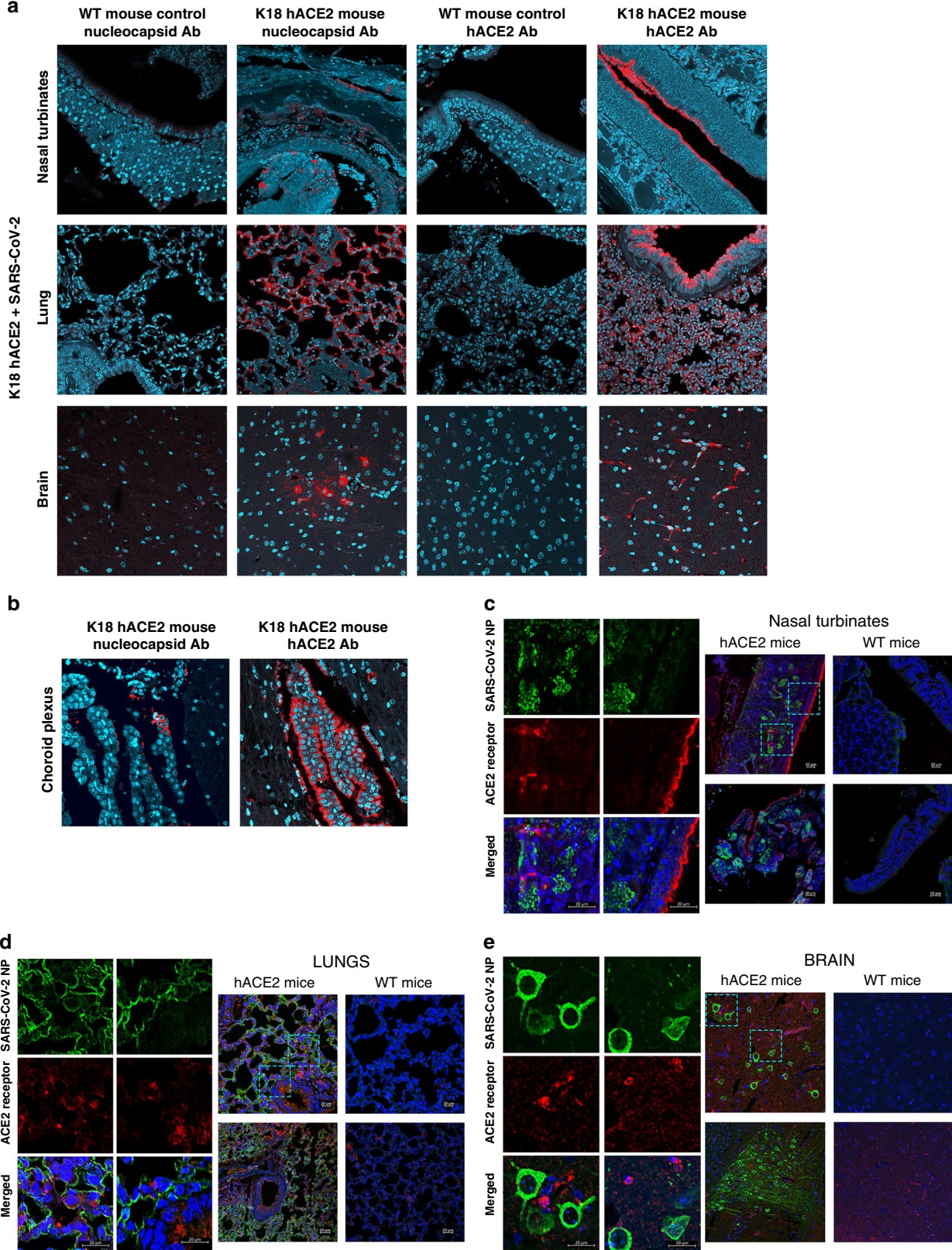

staining of NP using IHC, that colocalized with hACE2 in lungs, but was more focalized in the nasal turbinates and brain. Our IHC double staining in the brain also identified a specific population of pyramidal neurons in the cerebral cortex as a potential niche for SARS-CoV-2. K18 hACE2 transgenic mice developed progressive pneumonia by 4 DPI, which was accentuated at

6 DPI, there was evidence of intra-alveolar fibrin, cellular debris, vasculitis, and edema in the lung, likely driving morbidity and mortality. SARS-CoV-2 infection of K18 hACE2 transgenic mice was also associated by a marked significant increase in chemokine and cytokine production in the lung and spleen by 2 DPI. Although most chemokines and cytokines expression levels were

**Fig. 8 IHC examination of tissue from K18 hACE2 transgenic and WT C57BL/6 mice infected with SARS-CoV-2. a** WT and K18 hACE2 transgenic C57BL/6 mice nasal turbinate (top, at 2 DPI), lung tissue (middle, at 2 DPI), and brain (bottom, at 4 DPI) stained with an antibody against SARS-CoV-2 NP (red) or with antibody recognizing hACE2 receptor (red). **b** Infected choroid plexus in the brains of SARS-CoV-2-infected K18 hACE2 transgenic mice at 4 DPI. Left panel show that the hACE2 receptor (red) is highly expressed in the choroid plexus (all nuclei shown in aqua-blue). Right panel shows that some cells in the choroid plexus are infected with SARS-CoV-2 as these are positive to a SARS-CoV-2 NP (in red). **c** Nasal turbinates, **d** lung, and **e** brain from SARS-CoV-2-infected K18 hACE2 transgenic and wild-type mice at 2 DPI (bottom) and 4 DPI (top). Representative images of $n = 8$ for K18 hACE2 transgenic mice (50:50 male:female, per time point) and $n = 2$ for WT mice (50:50 male:female, per time-point) randomly chosen. Images magnified ×10 and ×20. Further magnification of selected areas (left, magnification bar 20 μm). hACE2 receptor (red), SARS-CoV-2 NP (green) as observed in nasal turbinates, lung, and brain tissues of K18 hACE2 transgenic mice (Blue: DAPI). Colocalization of hACE2 and SARS-CoV-2 NP is shown in the merged images. Data are representative over two independent experiments.

reduced by 4 DPI, MCP-1/CCL2, and IFN-λ remained elevated in the lung and virus persisted.

An important distinction between our study and others is that K18 hACE2 transgenic mice succumbed to SARS-CoV-2 infection by 6 DPI when infected at $1 \times 10^5$ PFUs. This lethal phenotype is not observed in other hACE2 mouse studies of SARS-CoV-2 infection, where ACE2 is expressed under a different promoter[12–19,28], including a study using ~$10^5$ PFU (i.n.) of the Hong Kong/VM20001061/2020 SARS-CoV-2 strain[12]. However, our results are supported by other unpublished and recently published studies using K18 hACE2 transgenic mice[29,30]. Some of these studies where less robust than ours based on the limited number of K18 hACE2 mice used, in some instances two per time/point, but showed similar lethality using $10^4$ to $10^5$ PFU of SARS-CoV-2 as inoculum[29,30].

K18 hACE2 transgenic mice express the hACE2 protein under the human K18 promoter, which induces high transgene expression specifically in airway epithelial cells[11,31]. The K18 hACE2 transgenic mice contain 2.5 kb of the K18 genomic sequence, including the promoter, first intron, and a translational enhancer (TE) sequence from alfalfa mosaic virus upstream of hACE2, followed by exons 6–7 and the poly(A) signal of the human K18 gene. This complete 2.5 kb K18 genomic sequence is necessary for the K18 hACE2 transgene expression in lung airway epithelium, and other organs such as the liver, kidney, and gastrointestinal tract[11,32]. This contrasts with mice expressing hACE2 under the mouse ACE2 promoter[14,16,19,33]. Thus, a potential difference in results between our study and others is the high expression levels of hACE2 in the airway epithelia in the K18 hACE2 transgenic mouse model used in this study. This concept is further supported by our IHC data showing higher levels of hACE2 expression in the airway epithelium (nasal turbinate and lung, Fig. 8) in K18 hACE2 transgenic mice. Of interest, other studies infected the K18 hACE2 transgenic mice with a lower MOI ($10^4$ PFUs) than used in this present study, wherein the majority of mice showed morbidity by 7 DPI but not mortality[12,13]. However, when K18 hACE2 transgenic mice were infected with $10^5$ PFU in those studies, they developed progressive weight loss and lung pathologies at 2 DPI and central nervous system (CNS) involvement by 6 DPI, 2-days later than our observations[28,32,34]. Thus, in addition to the presence of hACE2 on the upper and lower respiratory tract epithelium for establishing SARS-CoV-2 infection in K18 hACE2 transgenic mice, it appears that the number of viral particles during the initial exposure is also a critical determinant of developing severe COVID-19 morbidity and mortality in this model, similar to the situation with other respiratory viral infections (e.g., influenza)[35].

The term cytokine storm was first used during avian H5N1 and 1918 influenza virus infections, for SARS-CoV-1 and, more recently, for SARS-CoV-2[25,36–38]. In K18 hACE2 transgenic mice, we observed correlations between cytokines, the presence of virus, SARS-CoV-2 NP antigen, and pathogenesis (histopathology). As observed in our study, as well as reported in mice expressing

hACE2 under a mouse or adenovirus promoter[12–19], it appears that the magnitude of cytokine and chemokine responses during SARS-CoV-2 infection is another important determinant of disease severity and this result may be applicable to predicting a person's clinical outcome after SARS-CoV-2 infection [asymptomatic, mild COVID-19 or severe COVID-19 Acute Respiratory Distress Syndrome (CARDS)]. K18 hACE2 transgenic mice expressed an early (2 DPI) and mixed cytokine profile that included pro-inflammatory, TH1, TH2, and TH17 cytokines indicative of cytokine dysregulation in the lungs. This cytokine storm resolved by 4 DPI, even though mice were reaching a moribund state, with the exception of TNF and the type I and III IFNs, which remained significantly increased. These results suggest that TNF and type I and III IFN's may be important drivers of disease progression.

Although the amount of chemokines decreased at 4 DPI, MCP-1/CCL2 remained elevated indicative of ongoing tissue damage and cellular recruitment. MCP-1/CCL2 and IP-10/CXCL10 were produced at high levels in K18 hACE2 transgenic mice after SARS-CoV-2 infection, both, locally and systematically at 2 DPI, as well as in the brain at 4 DPI. Elevated MCP-1/CCL2 and IP-10/CXCL10 likely explained the accumulation of inflammatory cells in the lung including the observed neutrophils and monocytes (Table 1), and development of vasculitis that likely contributed to morbidity and mortality. In this context, gene expression signatures in the lungs of SARS-CoV-2-infected patients who succumb to viral infection showed high expression of MCP-1/CCL2 and IP-10/CXCL10, which was linked to type I and II IFN responses[36,39]. Early reports from COVID-19 patients also indicates that chemokines are elevated[25] and MCP-1/CCL2 and IP-10/CXCL10 have been implicated in COVID-19 patients[25], demonstrating that K18 hACE2 transgenic mice replicate cytokine and chemokine storm traits observed in humans, including recent reports of cytokine and chemokine production in human bronchoalveolar lavage fluid[40–42].

The consequences of reduced cytokine production at 4 DPI is unknown but we can speculate on several modes of action. This process could be driven by accumulation of lymphoid tissue around the airways, which dampens initial responses in severe influenza virus infection[43]. Alternatively, there may be a reestablishment of the balance between pro-inflammatory cytokines and their soluble receptors or inhibitors in the alveolar compartment[44]. The production of TH2 cytokines also represents another mechanism involved in regulating pro-inflammatory responses. The observed production of IL-10, although commonly recognized as an anti-inflammatory cytokine, plays a role in fibrosis, inducing collagen production and fibrocyte recruitment into the lung[36,45]. In this context, several correlative cytokine clusters linked by IL-10 were observed in multiple organs including the lungs and the brain. Elevated IL-10 can be associated with immune paralysis[36], altering the function of neutrophils and monocytes reaching the local infection site, potentially driving the expansion of the SARS-CoV-2 infection in

the K18 hACE2 transgenic mouse model. Alternatively, the high levels of IL-6 observed in tissues may provide a mechanism for enhancing TNF activity during acute viral infection[44] as well as B-cell antibody production and fibrosis[46]. Thus, it is conceivable that if one or more of these regulatory mechanisms are aberrantly regulated or absent, the pro- and anti-inflammatory balance critical for lung immune homeostasis might be disrupted, contributing to the cytokine storm observed in the K18 hACE2 transgenic mouse model upon SARS-CoV-2 infection. Interestingly, IL-4/IL-13 activation of the STAT-6 signaling pathway is associated with some lung pathologies including asthma[47,48]. The IL-6/IL-10/IL-13 correlation has also been described in patients with viral infections, where their expression levels in serum correlates with the disease clinical state[49], as well as to aortic aneurysm[50]. Indeed, influenza infection is related to heart pathologies[51].

Although SARS-CoV-2 was detected in the brain at 4 DPI, minimal histopathological changes were observed for the majority of mice, except for two mice that presented with multifocal perivascular neutrophils, lymphocytes, microglial cells and necrotic debris, vasculitis, and inflammation which extended to the meninges with hemorrhage. Brain cytokine responses were dominated by TH2 cytokines (IL-4, IL-13, IL-10, IL-27, and IL-33) which may reflect the natural TH2 status of the brain[52,53]. IL-27 is known to regulate inflammation in the CNS through upregulation of IL-10[54–56], but can also potentiate inflammation through induction of IL-12[57]. The cytokine profiles observed in SARS-CoV-2-infected mice suggest it is the former response at play, working to counterbalance the potent peripheral TH1 responses. We also observed increases in cytokine production at 2- and 4 DPI that included a robust IL-1β response. The detection of IL-1β in brain at 2 DPI suggests that virus spread to brain was delayed relative to lung and spleen[25]. Interferon type II and III were significantly lower in the brain of SARS-CoV-2-infected K18 hACE2 transgenic mice when compared to WT C57BL/6 mice at all-time points studied, and interferon type I was not detected, which could be a deciding factor driving colonization of the brain by SARS-CoV-2 in this mouse model. Indeed, deviation from homeostatic IFN type I/IFN type II balance contributes to insufficient immune surveillance of the CNS and loss of immune-dependent protection, or immune-mediated destruction[58].

Most intriguing was that many TH2 cytokines (IL-1β, IL-27, IL-4, IL-13, IL-33, and IL-10) were also elevated in the brain of WT C57BL/6 control mice at 2 and 4 DPI. Although we did not find any virus in WT C57BL/6 mice, it suggests that the virus might be privy to the blood–brain barrier because the cytokine pattern in the brain was similar to K18 hACE2 transgenic mice. C57BL/6 mice express the mouse ACE2 protein and studies have shown that ACE2 gene-disrupted mice are less susceptible to SARS-CoV-1[59]. Therefore, it is possible that endogenous ACE2 in mice could drive a cytokine response in the brain of WT C57BL/6 mice infected with SARS-CoV-2 while showing very limited peripheral responses. Overall, these data suggest that there may be some tropism for SARS-CoV-2 in the brain of WT mice, supporting some reports of neurological issues in humans with COVID-19[60]. We also observed sex differences in K18 hACE2 transgenic mice with males exhibiting a TH1 and TH17 phenotype at 4 DPI, whereas female K18 hACE2 transgenic mice had a TH2 responses at the same time point, similar to studies of acute LPS-induced inflammation[26,27], which suggests that males may be more prone to exhibit neurological problems during SARS-CoV-2 infection, independent of viral load.

Although our findings clearly identify the K18 hACE2 transgenic mouse as a model for SARS-CoV-2 infection and COVID-19 disease, our studies have some limitations. The primary limitation was the number of mice that were available for study,

owing to limited supply from the vendor. We maximized our studies by performing two independent experiments that cross-validated each other in survival and pathogenesis results. Both studies showed considerable sensitivity of K18 hACE2 transgenic mice to SARS-CoV-2 infection. We chose a relatively high viral dose (MOI $1 \times 10^5$ PFU/mouse) that was capable of causing extensive disease, but we were unable to include additional time points to further understand the time course of immune responses (e.g., 1 DPI) or whether the chemokine and cytokine storm increased again at study endpoints (e.g., 5/6 DPI). Likewise, the limited number of K18 hACE2 transgenic mice did not allow us to use different viral doses to calculate the mouse lethal dose 50 ($MLD_{50}$) of SARS-CoV-2. Inclusion of WT C57BL/6 control mice in our studies confirmed that mice are not naturally susceptible to infection with SARS-CoV-2 without experimental manipulation to express hACE2, further confirming that hACE2 is the receptor for SARS-CoV-2[61–64]. The presence of pathology in the liver and intestine without detectable viral load may simply reflect the detection limit of our PFU assay, but it is also possible that hepatic and intestinal changes were a consequence of circulating inflammation or gut leakage of microbial microflora that would lead to LPS-induced responses in tissues.

Several groups are developing large animal models of SARS-CoV-2 infection and COVID-19, with a primary focus on non-human primates (NHP) models. Reports indicate that macaques develop self-limiting disease with a strong anamnestic responses (innate and adaptive)[65–67], whereas vervets may model signs of acute respiratory distress[68]. These models are necessary for pre-clinical testing for safety, immunogenicity, and efficacy of vaccines and therapeutics. However, no NHP models display the end-stage COVID-19 outcomes that are effectively captured in the K18 ACE2 transgenic mice. Hence, evaluation of therapeutics and antivirals must also leverage transgenic murine models of end-stage lethal COVID-19 disease, characterized by ARDS. The K18 hACE2 mouse model can work in concert with other rodent models, such as the Golden Syrian hamster model[69–71], to provide two-step testing of therapeutics. This would fill a critical gap and relieve the pressure on the limit number of NHPs available for this purpose worldwide. Moreover, the K18 hACE2 mouse model could also be used to assess the contribution of host cellular factors in SARS-CoV-2 infection or associated COVID-19 disease, for example, through crossing K18 hACE2 mice with IFNAR KO mice to assess the contribution of type I IFN in SARS-CoV2 infection[72,73].

Altogether, our study defines the K18 hACE2 transgenic mouse as an important small animal model to evaluate SARS-CoV-2 pathogenicity and to assess protection efficacy of prophylactic (vaccines) and therapeutic (antivirals) approaches against SARS-CoV-2 infection with specific readouts such as morbidity, mortality, cytokine and chemokine storms, histopathology, and viral replication.

## Methods

**Ethics statement**. All experimental procedures with animals were approved by the Texas Biomedical Research Institute (Texas Biomed) Institutional Biosafety Committee (IBC, #20-004 and #20-010) and Institutional Animal Care and Use Committee (IACUC, #1708 MU) and under Biosafety Level 3 (BSL-3) and animal BSL3 (Animal Biosafety Level (ABSL)-3) facilities at Texas Biomed.

**Virus, cells, and viral propagation**. SARS-CoV-2, USA-WA1/2020 strain (Gen Bank: MN985325.1), was obtained from BEI Resources (NR-52281). SARS-CoV-2 USA-WA1/2020 was isolated from an oropharyngeal swab from a patient with a respiratory illness in January 2020 in Washington, US. The virus stock obtained from BEI Resources was a passage (P) 4 stock, and was used to generate a master P5 seed stock. The P5 stock was used to generate a P6 working stock. P5 and P6 stocks of SARS-CoV-2 were generated by infecting at low multiplicity of infection (MOI, 0.001) Vero E6 cells obtained from the American Type Culture Collection (ATCC, CRL-1586). At 72 h post infection, tissue culture supernatants

were collected and clarified before being aliquoted and stored at −80 °C. Standard plaque assays (PFU/ml) in Vero E6 cells were used to titrate P5 ($1.7 \times 10^6$ PFU/ml) and P6 ($2.6 \times 10^6$ PFU/ml) viral stocks. P5 seed and P6 working stocks were full sequenced using next generation sequencing (NGS) and were 100% identical to the BEI original P4 stock, without deletions or mutations compromising virus infectivity. We examined both our viral master seed and working stocks for single nucleotide variants with a particular emphasis on those impacting the furin-like cleavage site, which has been described to influence the ability of SARS-CoV-2 to transmit between species. Our results confirmed that the 24nt Bristol deletion, or other deletions/mutations affecting the furin cleavage site, were not present in our viral stocks[74]. The virus used in this study, SARS-CoV-2 was fully sequenced using NGS with the accession deposit number MT576563.

**Mice**. Specific-pathogen-free, 4–5-weeks-old, female and male B6.Cg-Tg(K18-ACE2)2Prlmn/J (Stock No: 034860, K18 hACE2) hemizygotes, or wild-type (WT) C57BL/6 control mice, were purchased from The Jackson Laboratory (Bar Harbor, ME). K18 hACE2 transgenic and WT C57BL/6 mice were identically maintained in micro-isolator cages at ABSL-2 for noninfectious studies, or at ABSL-3 for studies involving SARS-CoV-2. Mice were provided sterile water and chow ad libitum and acclimatized for at least one week prior to experimental manipulation.

Based on the limited number of K18 hACE2 transgenic mice from The Jackson Laboratory, $n = 7$ each for female and male mice (14 total) were used for morbidity and mortality studies, whereas $n = 7$ each for female and male mice (14 total) were used for viral titers at 2 and 4 DPI, and $n = 3$ each for female and male (six total) at 6 DPI were used for viral titers, respectively. An $n = 3$ each for female and male K18 hACE2 transgenic mice (six total) were used as mock-infected controls in the morbidity and mortality studies, and one female and one male mock infected for the viral determination studies. For C57BL/6 WT (The Jackson Laboratory), $n = 4$ each for female and male mice (eight total) infected and $n = 3$ each for female and male mice (six total) mock-infected were used for morbidity and mortality, whereas $n = 4$ each for female and male mice (eight total) infected were used for viral titers at 2 and 4 DPI, and one female and one male mock infected for viral determination.

**Mouse infection and sample processing**. K18 hACE2 transgenic and WT C57BL/6 mice were either mock (PBS)-infected (controls) or infected intranasally (i.n.) with $1 \times 10^5$ PFU of SARS-CoV-2 in a final volume of 50 μl following iso-flurane sedation. Limited numbers of available K18 hACE2 transgenic mice reduced the study to a single exposure dose of SARS-CoV-2. After viral infection, mice were monitored daily for morbidity (body weight) and mortality (survival). Mice showing >25% loss of their initial body weight were defined as reaching experimental end-point and humanely killed.

In parallel, K18 hACE2 transgenic or WT C57BL/6 mice were infected and euthanized at 2, 4, or 6 DPI. For viral titers, chemokine/cytokine, histopathology and IHC analyses, seven female and seven male for 2 and 4 DPI, and three female and three male for 6 DPI SARS-CoV-2-infected; and one female and one male mock-infected K18 hACE2 mice were used. As controls for these analyses, four female and four male SARS-CoV-2-infected for 2 and for 4 DPI, and one male and one female for mock WT C57BL/6 mice were used. Ten tissues (nasal turbinate, trachea, lung, heart, kidney, liver, spleen, small intestine, large intestine, and brain) were harvested from each mouse. Half organ was fixed in 10% neutral buffered formalin solution for molecular pathology analyses and the other half was homogenized in 1 mL of PBS using a Precellys tissue homogenizer (Bertin Instruments) for viral titration. Tissue homogenates were centrifuged at $21,500 \times g$ for 5 min and supernatants were collected for measurement of viral load and chemokine/cytokine analyses.

**Measurement of viral loads**. Confluent monolayers of Vero E6 cells (96-well plate format, $4 \times 10^4$ cells/well, duplicate) were infected with 10-fold serial dilutions of supernatants obtained from the organ homogenates. Virus was adsorbed for 1 h at 37 °C in a humidified 5% $CO_2$ incubator. After viral adsorption, cells were washed with PBS and incubated in post-infection media containing 1% microcrystalline cellulose (Avicel, Sigma-Aldrich). Infected cells were incubated in a humidified 5% $CO_2$ incubator at 37 °C for 24 h. After viral infection, plates were inactivated in 10% neutral buffered formalin (ThermoFisher Scientific) for 24 h. For immunostaining, cells were washed three times with PBS and permeabilized with 0.5% Triton X-100 for 10 min at room temperature. Cells were then blocked with 2.5% bovine serum albumin (BSA) in PBS for 1 h at 37 °C, followed by incubation with 1 μg/ml of a SARS-CoV-1 NP cross-reactive monoclonal antibody (MAb), 1C7, diluted in 1% BSA for 1 h at 37 °C. After incubation with the primary NP MAb, cells were washed three times with PBS, and developed with the Vectastain ABC kit and DAB Peroxidase Substrate kit (Vector Laboratory, Inc., CA, USA) according to the manufacturers' instructions. Viral determinations were analyzed using C.T.L. Immunospot v7.0.15.0 Professional Analysis DC. Viral titers were calculated as PFU/mL.

**Multiplex cytokine assay**. Cytokines and pro-inflammatory markers were measured using a custom 18-multiplex panel mouse magnetic bead Luminex assay

(R&D Systems, Mouse 18-Plex, lot L134111), following the manufacturer's instructions. Immunoassays were performed in the ABSL-3 and samples decontaminated by an overnight incubation in 1% formalin solution before readout on a Luminex 100/200 System running on Xponent v4.2 with the following parameters: gate 8000–16,500, 50 μl of sample volume, 50–100 events per bead, sample timeout 60 s, low PMT (LMX100/200: Default). Acquired data were analyzed using Millipore Sigma Belysa™ v1.0.

**Interferon (IFN) ELISA**. Mouse IFN-α (Type I) and IFN-λ (Type III) were measured by enzyme-linked immunosorbent assays (ELISA) (PBL Assay Science) following the manufacturer's recommendations, detecting all 14 known IFN-α subtypes and IFN-λ 2 and 3. IFNs Type I and III were measured on a GloMax instrument running Glomax Explorer software v3.0.

**Histopathology analyses**. Tissues were fixed in 10% neutral buffered formalin, embedded in paraffin blocks, and sectioned at 4 μm thickness. Sections were stained with Haemotoxylin and Eosin (H&E) and evaluated using light microscopy in a blinded manner by a board certified veterinary pathologist. For histopathology, images were acquired using an Olympus Bx46 microscope and Olympus CellSense v1.18 Life science Imaging software. For CD3 and CD20 labeling, tissue sections were cut at four microns, deparaffinized, and antigen retrieval performed using a heat induced epitope retrieval (HIER) method. Tissue sections were stained with CD3 (Dako/Agilent, Cat. #A0452) and CD20 (Abcam, Cat. #ab64088) using Dako's Envison + System, HRP. Table 1 was generated from C57BL/6 WT $n = 8$ (four males and four females) for each time point 2 and 4 DPI, and K18 hACE2 transgenic $n = 8$ (four males and four females) for each time point 2 and 4 DPI, for a total of 32 mice. Table 2 was generated from K18 hACE2 transgenic mice only at 6 DPI $n = 6$ (three males and three females). Table 3 and Supplementary Table 1 were generated from three time points 2, 4, and 6 DPI, $n = 6$ for each group/time-point. All animals were randomly chosen.

**Immunohistochemistry assays**. For Immunostaining and confocal microscopy for hACE2 and SARS-CoV-2 NP[75,76], 5 μm tissues sections were mounted on Super-frost Plus Microscope slides, baked overnight at 56 °C and passed through Xylene, graded ethanol, and double distilled water to remove paraffin and rehydrate tissue sections. A microwave was used for HIER. Slides were boiled for 20 min in a TRIS-based solution, pH 9.0 (Vector Labs H-3301), containing 0.01% Tween-20. Slides were briefly rinsed in distilled hot water, and transferred to a hot citrate based solution, pH 6.0 (Vector Labs H-3300) and cooled to room temperature. Once cool, slides were rinsed in TRIS buffered saline (TBS) and placed in a black, humidifying chamber and incubated with Background Punisher (Biocare Medical BP974H) for 40 minutes. Subsequently, slides were stained with primary rabbit antibodies against the following proteins: SARS-CoV-1 NP at 1:4000 dilution (a rabbit polyclonal antibody shown to cross-react with SARS-CoV-2 NP), hACE2 receptor (hACE2 Recombinant Rabbit Monoclonal Antibody, 1:50, ThermoFisher, Cat#SN0754) and DAPI nuclear stain (1:20,000, Invitrogen, Carlsbad, CA). The first primary antibody, hACE2, was detected with goat anti-rabbit-AP developed with Permanent Red warp (mach3 Biocare Medical). After a second round of HIER with citrate buffer, the SARS-CoV-1 NP primary antibody was applied overnight and after several washes, a secondary antibody (goat anti-rabbit-Alexa 488, ThermoFisher, Cat#A-11008) was applied at 1:1000 for an hour at room temperature. After washing the excess of antibody, the slides were incubated with DAPI for nuclear stain (1:20,000, Invitrogen, Carlsbad, CA). Confocal microscopy images were acquired using Zeiss LSM 800 microscope running Zen Blue Edtion v2.1 software and the same software was used for any scale bars presented and image processing.

**Statistical analysis**. Statistical significance was determined using Prism v8.0 software (GraphPad Software, San Diego, CA). The unpaired, two-tailed Student's $t$ test was used for two group comparisons for each time-point and reported as $p < 0.05$; $p < 0.005$; $p < 0.0005$. For individual cytokine/chemokine analyses, correlations were made between analyte concentrations [measured as the area under the curve (AUC) or the peak concentration] and viremia using linear regression by hierarchically clustered Pearson test. For the correlation data, RStudio v1.2.5033 running R v4.0.2 and packages (ggplot2) v3.3.2 and (ggcorrplot) v0.1.3 was used. Multiple comparisons among groups and/or time-points were analyzed using one-way analysis of variance with Tukey's post test and reported as $p < 0.05$; $p < 0.005$; $p < 0.0005$.

**Reporting summary**. Further information on research design is available in the Nature Research Reporting Summary linked to this article.

## Data availability

The authors declare that all data supporting the findings of this study are available within the paper and its Supplementary Information files and from the corresponding authors upon reasonable request. The virus used in this study was fully sequenced using next generation sequencing (NGS) with the GenBank accession deposit number MT576563. Source data are provided with this paper.

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

## Acknowledgements

These studies were supported by philanthropic donations. We thank Dr. Abul Azad for editing, Dr. Marcel Daadi for identifying the type of neuron being infected by SARS-CoV-2 in the K18 hACE2 transgenic mouse model, and Drs. Cat Lutz and Steve Rockwood for facilitating the access to The Jackson Labs K18 hACE2 transgenic mice to conduct this study.

## Author contributions

Planned, executed studies, and data analyses (F.S.O., J.-G.P., P.A.P., O.G., A.A., A.A.G., A.O.F., S.G., A.G.V., C.Y., K.C., C.H., V.D.L.P., L.M.P., K.J.A., H.M.S., A.S., J.I.G., A.W., R.N.P., M.G., J.M., C.C., S.E., O.H.R., S.D.M., K.N.K., R.E., S.H.U., X.A., J.T., L.M.S., and J.B.T.); edited the manuscript (O.G., C.R.A.H., C.-Christi, J.L.P., T.A., R.C.Jr., L.D.G., E.J. D.Jr., D.K., and L.S.S.); wrote the manuscript (J.T., L.M.S., and J.B.T.).

## Competing interests

The authors declare no competing interests.
