## [Peer Review File · Nature Communications]

REVIEWER COMMENTS

Reviewer #1 (Remarks to the Author):

The manuscript by Fatai S. Oladunni and co-authors investigated if a previously developed K18-hACE2 transgenic mouse is susceptible to SARS-CoV-2 infection. The authors intranasally infected both WT C57 mice and K18-hACE2 mice with SARS-CoV-2 and monitored mice for morbidity and mortality. Notably, all infected K18 mice succumbed to viral infection by 6 DPI. In addition, authors evaluated the viral load in the nasal turbinates, lungs and brains. Further, they reported K18 mice generate diseases that partly simulate human infection by SARS-CoV-2, such as developing interstitial pneumonia and local/systemic chemokine/cytokine storm. Overall, the authors defined the K18 mice as a valuable rodent animal model to evaluate SARS-CoV-2 pathogenesis and to assess prophylactic or therapeutic treatments.

Although there were several transgenic/KI mouse models have been reported, all of them were non-lethal or sub-lethal infection by SARS-CoV-2. This work introduced a gender independent lethal SARS-CoV-2 infection model which is important for evaluating vaccines and antivirals. There are some comments for the authors' consideration:

- 1) Chemokine/cytokine storm were mostly detected in peripheral blood after human infection, is the chemokine/cytokine data in infected mouse peripheral blood available?
- 2) Authors mentioned "Viral NP antigen doesn't co-localize in the same regions as hACE2 except in the lung (Fig. 6A and data not shown), can authors present NP and hACE2 staining in one section that may clearly show the correlation between SARS-CoV-2 infection and hACE2 expression?"
- 3) In Fig. 1D.E (also in Fig. S2), virus can not be detected in some mouse lung and brain by 4-DPI, are the missing point in lung and brain from the same individual? What the possible explanation that virus was absent in two mice lung/brain at 4-DPI while all infected K18 mice were succumbed at 6-DPI in survival group?
- 4) Virus load and relevant pathology changes in dead mouse tissues should be provided.
- 5) Only one male individual showed high viral load in nasal turbinate (much higher than lung), dose it correlated with symptoms and/or chemokine/cytokine level?
- 6) What's the possible physiological significance of hepatic and intestinal pathology change while virus can't be detected there.
- 7) As authors mentioned "other studies infected the K18 hACE2 transgenic mice with a lower MOI (104 PFUs) than used in this present study, wherein the majority of mice showed morbidity by 7-DPI but not mortality." in discussion, further test for LD50 of K18 mice will be of great use for antiviral test in the future.

8) Dysregulated chemokine/cytokine lead to tissue damage, did the authors characterize the leukocyte population/function change in lung? which can better explain the significance of correlated chemokine/cytokine clusters.

Reviewer #2 (Remarks to the Author):

K18 ACE2 mouse is good model which can lead to death and loss of weight after SARS-CoV-2 infection. Oladunni et al. systematically analyzed its pathogenesis and immune response. However, three papers (one published in Am J Trop Med Hyg, two preprint in bioRxiv) related to K18 ACE2 mouse were published. Compare to those papers, no novel finding was reported in this manuscript. Another problem is that the number of mice used in this study was small and the results may not be representative. For eacmples, SARS-CoV-2 RNA were detected in Brain. But other reports showed that only small percentage of brains was positive for SARS CoV 2 RNA.

Other comments:

1. The first sentence in Abstract “Vaccine and antiviral development against SARS-CoV-2 infection or COVID-19 disease currently lacks a validated small animal model” is not objective. So far, some mouse models have been used in products development and published.
2. It is better to provide more information related to sequence of SARS CoV 2 used in this study. So readers can know whether this strain is mutant or not.
3. Although SARS CoV 2 was detected in nasal turbinate, lung and brain, the viral titer was quite low.

Response to Reviewers' Comments

Reviewer #1

Introductory comments: Although there were several transgenic/KO mouse models have been reported, all of them were non-lethal or sub lethal infection by SARS-CoV-2. This work introduced a gender independent lethal SARS-CoV 2 infection model which is important for evaluating vaccines and antivirals.

Response: We appreciate the reviewer's overall positive comments and recognition of the novelty of our findings with regard to lethality of the K18 hACE2 transgenic mice, and the importance of our sex independence findings.

1) Chemokine/cytokine storm were mostly detected in peripheral blood after human infection, is the chemokine/cytokine data in infected mouse peripheral blood available?

Response: Unfortunately, we are unable to provide this data because blood was not collected at the time of necropsy. However, there is increasing evidence from humans that cytokines and chemokines are elevated in the BAL, which supports our observations (References 44-46 in the manuscript). We have included additional text to acknowledge those new findings in humans (lines 540-544).

2) Authors mentioned "Viral NP antigen doesn't co-localize in the same regions as hACE2 except in the lung (Fig. 6A and data not shown), can authors present NP and hACE2 staining in one section that may clearly show the correlation between SARS-CoV-2 infection and hACE2 expression?"

Response: We concur with the comment made by the reviewer. We have included new figures that show dual staining of hACE2 and NP in tissues that clearly identifies the lack of co-localization between virus and hACE2 expression in the nasal turbinates and brain. Co-localization was only apparent in the lung. Our dual staining in the brain also uncovered a preferential pyramidal neuron cell type as a niche for SARS-CoV-2 infection in the K18 hACE2 mouse model (**Fig 8C**, lines 460-463).

3) In Fig. 1D.E (also in Fig. S2), virus cannot be detected in some mouse lung and brain by 4 DPI, are the missing point in lung and brain from the same individual? What the possible explanation that virus was absent in two mice lung/brain at 4-DPI while all infected K18 mice were succumbed at 6-DPI in survival group?

Response: The reviewer brought up an important point. To address the reviewer's concern, we have performed a replicate experimental infection for each time point for the morbidity and mortality curve, and the viral titers determination (new **Figs 1, S1, and S2**). Data have been amended. Our new data show that all mice except one had detectable viral titers in nasal turbinates, as well as in the lungs and brain. In the nasal turbinates the viral titers remained low (between 10^2 and 10^3) but were constant throughout the study; however, in lung the viral load

peaked at 2-DPI (10^5) and then decreased over time (10^2 by 6-DPI). This is contrary to what was observed in the brain, where the highest viral titers detected occurred at 6-DPI (10^7). As previously, we did not observe sex related differences in these new studies (**Fig S2**).

4) Virus load and relevant pathology changes in dead mouse tissues should be provided.

Response: We have included viral titers (**Figs 1 and S2**), histopathology (**Fig 6 and 7**), a summary of pathology (**Table 2**) and cytokine/chemokine correlations for 6-DPI in the brain. At 6-DPI the brain had significantly higher viral titer (**Fig 4C**) and mice succumbed to infection. Histopathology for multiple organs including brain at 6-DPI are discussed in the revised version of the manuscript (lines 429-438, Lines 476-478).

5) Only one male individual showed high viral load in nasal turbinate (much higher than lung), dose it correlated with symptoms and/or chemokine/cytokine level?

Response: We apologize for the lack of clarity in the original version of the manuscript. we have performed a replicate experimental infection for each time point and data have been amended. Please see response to point 3 above and new **Fig 1 and S2**.

6) What's the possible physiological significance of hepatic and intestinal pathology change while virus can't be detected there.

Response: The presence of pathology in the liver and intestine without detectable viral load may simply reflect the detection limit of the assay, but it is also possible that hepatic and intestinal changes were a consequence of circulating inflammation. We thank the reviewer for bringing up an important point. However, we do not have a valid scientific explanation for the hepatic and intestinal pathology changes at this time. We will hopefully be able to address this important comment in our future studies. We have added some text to reconcile this on lines 615-618.

7) As authors mentioned "other studies infected the K18 hACE2 transgenic mice with a lower MOI (104 PFUs) than used in this present study, wherein the majority of mice showed morbidity by 7-DPI but not mortality." in discussion, further test for LD50 of K18 mice will be of great use for antiviral test in the future.

Response: We agree with the comment made by the reviewer. Unfortunately, we had a limited number of K18 hACE2 transgenic mice which was insufficient to conduct MLD50 studies. We are planning to conduct these important studies in the near future since the K18 hACE2 mice are now more readily available from The Jackson Laboratory. Based on the comment made by the reviewer, we have included a comment in the discussion, at lines 611-612.

8) Dysregulated chemokine/cytokine lead to tissue damage, did the authors characterize the leukocyte population/function change in lung? which can better explain the significance of correlated chemokine/cytokine clusters.

Response: We include new data showing quantification of leukocytes (histiocytes, neutrophils, T and B cells) in the lung (**Fig 6**) and brain (**Fig 7**), and summarize these findings in **Table 3** and **Fig S9**. A discussion of these results has been added (lines 429-445). Due to the priority of lung tissues for viral load and histopathology we were unable to collect sufficient tissue to perform functional leucocyte studies but agree with the reviewer that this would be informative for follow up in future studies.

Reviewer #2

Introductory comments: However, three papers (one published in Am J Trop Med Hyg, two preprint in bioRxiv) related to K18 ACE2 mouse were published. Compare to those papers, no novel finding was reported in this manuscript.

Response: We respectfully disagree with the overall comment made by this reviewer. At the time of initial submission of our manuscript, only one manuscript was officially published which did not show lethality. Posting a paper in *BioRxiv* is not validation of peer-review for formal publication. Our manuscript was similarly posted to *BioRxiv* at the time of submission to Nature Communications (prior to others), and took 40 days in peer reviewed. We were unable to accelerate this process and it was inevitable that additional papers would be posted to *BioRxiv* over time. Since then and to the time of this resubmission, several publications using this model have been published, two of them showing lethality (one of them only using two mice per time-point for viral titer determination). We believe that our manuscript presents additional important findings that merit publication, and support other recently published studies to establish a lethal animal model to test new therapies and vaccines against SARS-CoV-2. Peer-reviewed papers published since submission and resubmission has been cited in the revised version of our manuscript (lines 68-69, lines 485-492)

Another problem is that the number of mice used in this study was small and the results may not be represented.

Response: We agree with the reviewer that the number of mice used in our study was low, a consequence of the very limited numbers provided by The Jackson laboratories to test this new model for SARS-CoV-2 infectivity. To increase robustness, we have since repeated our studies using additional mice (now commercially available, with 2-month wait list) and show that our findings are highly reproducible.

For examples, SARS-CoV-2 RNA were detected in Brain. But other reports showed that only small percentage of brains was positive for SARS CoV 2 RNA.

Response: In our studies, SARS-CoV-2 was detected in the brain of infected K18 hACE2 transgenic mice. In addition, our IHC studies clearly show a population of pyramidal neuron types as a potential niche for SARS-CoV-2 studies. We repeated these experiments and obtained similar results. We were able to detect SARS-CoV-2 in the brain of infected K18 hACE2 mice by 4-DPI and these viral titers increased until the mice succumbed to viral infection (6-DPI). We cannot reconcile our findings with other studies posted on *BioRxiv* but it is possible that differences are a consequence of infection doses, tissue collection or processing methods. We have repeated our studies and observed the presence of SARS-CoV-2 in the brain of K18 hACE2 transgenic mice at 4- and 6-DPI, further confirming our initial results. Additionally, similar detection of SARS-CoV-2 in the brain of K18 hACE2 mice has now been reported in manuscripts currently posted at *BioRxiv*:

<https://www.biorxiv.org/content/10.1101/2020.08.11.246314v1>

<https://www.biorxiv.org/content/10.1101/2020.08.07.242073v1>

<https://www.biorxiv.org/content/10.1101/2020.07.06.190066v1>

Other comments:

1. The first sentence in Abstract “Vaccine and antiviral development against SARS-CoV-2 infection or COVID-19 disease currently lacks a validated small animal model” is not objective. So far, some mouse models have been used in products development and published.

Response: We agree with the comment made by the reviewer. Based on the reviewer’s concern, we have modified the revised version of the manuscript text (lines 27-28).

2. It is better to provide more information related to sequence of SARS CoV 2 used in this study. So readers can know whether this strain is mutant or not.

Response: The SARS-CoV-2 strain used was obtained from BEI Resources. We received a passage 4 (P4) of the virus that was used for propagating and generating our own virus stocks (P5 and P6). Importantly, we have used next generation sequencing (NGS) to fully sequencing our viral stocks. Our NGS sequencing results demonstrate that our viral stocks did not have any mutations relative to the original stock provided by BEI resources. This information has been added to the Materials and Methods section (lines 121-128).

3. Although SARS CoV 2 was detected in nasal turbinate, lung and brain, the viral titer was quite low.

Response: We agree with the reviewer. Our results, that include a new replicate experiment, (**Fig 1** and **S2**) confirm that viral titers remained low but constant (between 10^2 and 10^3) in nasal turbinates. However, viral load in the lung peaked at 2-DPI (10^5) and decrease overtime by 6-DPI (10^2). This is contrary to what was observed in the brain, where the highest viral titers were detected at 6-DPI (10^7). As previously, we did not observe any sex related differences in our new studies (**Fig S2**).

REVIEWER COMMENTS

Reviewer #1 (Remarks to the Author):

Authors have made extensive revision and the manuscript is greatly improved.

Minor revision:

1. Fig. 8C, this reviewer observed in the figure that there is co-localization of ACE2 and SARS-CoV-2 NP in nasal turbinate and brain, but much less than in lung. Authors please double-check.
2. Fig. S1 C, the drawing is missing.

Reviewer #2 (Remarks to the Author):

All responses are reasonable. I have no other comments.

Reviewer #1 (Remarks to the Author):

Comment: Authors have made extensive revision and the manuscript is greatly improved.

Response: *We thank the reviewer's constructive review and positive comments about our revised manuscript.*

Minor revision:

Comment 1: Fig. 8C, this reviewer observed in the figure that there is co-localization of ACE2 and SARS-CoV-2 NP in nasal turbinate and brain, but much less than in lung. Authors please double-check.

Response: *We agree with the comment made by the reviewer that there is some co-localization of hACE2 and SARS-CoV-2 NP in nasal turbinates and brain, although not at the same extent observed in the lungs. This has been changed in the text (see lines 458-460, 469-473, and 979-983 in the revised manuscript). In addition, we redid **Fig 8** showing the presence of hACE2 (red), SARS-CoV-2 NP (green) and both together (yellow) (see new **Fig 8C-E**). Magnifications in the nasal turbinates, lung and brain are also provided to make this point clear.*

Comment 2: Fig. S1 C, the drawing is missing.

Response: *We thank the reviewer for bringing this editorial deficiency and apologize for this oversight. This information has now been added into **supplemental Fig S1**.*

Reviewer #2 (Remarks to the Author):

Comment: All responses are reasonable. I have no other comments.

Response: *We also appreciate reviewer #2 comment, indicating that we favorable addressed her/his previous concerns.*